# INNOGYM: BENCHMARKING THE INNOVATION POTENTIAL OF AI AGENTS

**Jintian Zhang**♠♡ , **Kewei Xu**♠ , **Jingsheng Zheng**♠♡, **Zhuoyun Yu**♠,
**Yuqi Zhu**♠♡, **Yujie Luo**♠♡, **Lanning Wei**♣♡, **Shuofei Qiao**♠, **Lun Du**♣♡,
**Da Zheng**♣♡, **Shumin Deng**◇, **Huajun Chen**♠♡, **Ningyu Zhang**♠♡†
♠Zhejiang University ♣Ant Group ◇National University of Singapore
♡Zhejiang University - Ant Group Joint Laboratory of Knowledge Graph
{zhangjintian,zhangningyu}@zju.edu.cn, zhengda.zheng@antgroup.com
⌂ https://github.com/zjunlp/igym

## ABSTRACT

LLMs and Agents have achieved impressive progress in code generation, mathematical reasoning, and scientific discovery. However, existing benchmarks primarily measure correctness, overlooking the diversity of methods behind solutions. True innovation depends not only on producing correct answers but also on the originality of the approach. We present **InnoGym**, the first benchmark and framework designed to systematically evaluate the innovation potential of AI agents. InnoGym introduces two complementary metrics: performance gain, which measures improvement over the best-known solutions, and novelty, which captures methodological differences from prior approaches. The benchmark includes 18 carefully curated tasks from real-world engineering and scientific domains, each standardized through resource filtering, evaluator validation, and solution collection. In addition, we provide **iGym**, a unified execution environment for reproducible and long-horizon evaluations. Extensive experiments show that while some agents produce novel approaches, their lack of robustness limits performance gains. These results highlight a key gap between creativity and effectiveness, underscoring the need for benchmarks that evaluate both.

## 1 INTRODUCTION

In recent years, LLMs (Jaech et al., 2024; DeepSeek-AI et al., 2025a) and Agents (Wang et al., 2024; Guo et al., 2024) have made rapid progress in areas such as code generation (Jain et al., 2025; Chen et al., 2021), mathematical reasoning (Hendrycks et al., 2021; Cobbe et al., 2021), and scientific discovery (Majumder et al., 2025; Jing et al., 2025). However, most existing benchmarks focus solely on whether an answer is correct. Under this paradigm, any output that passes test cases or matches the reference answer is deemed successful. Yet intelligence and innovation lie not only in the *results*, but also in the *methods*: two agents may arrive at the same correct answer while following entirely different approaches. Such methodological differences are often overlooked in current evaluation frameworks.

To address this gap, we propose a framework for evaluating innovation that formalizes each task as a quadruple $(P, S, V, D)$. Here, $P$ denotes the problem instance, $S$ the solution space, $V$ the performance measure, and $D$ the measure of dissimilarity between solutions. On top of this formulation, we introduce two key metrics: Performance gain $G$ and Novelty $N$. Performance gain $G$ quantifies the improvement of a solution relative to the best-known baseline, while Novelty $N$ captures the methodological difference between a new solution and prior ones. Together, these metrics enable us to assess both *performance breakthroughs* and *methodological innovation*.

Building on this framework, we present **InnoGym**, which consists of two complementary components: **iBench** and **iGym**. iBench is the first benchmark specifically designed to evaluate the innovation potential of AI agents. It includes 18 carefully curated *Improvable Tasks*, selected from

---

†Corresponding author.

real-world engineering (e.g., ROADEF Challenge[1]) and scientific problems (e.g., 2D-BPP (Chung et al., 1982)) where there remains clear room for improvement in both performance and methodology. These tasks stand in contrast to solved problems (with no remaining improvement margin) and exploratory problems (lacking human baselines or reliable validation). To ensure fairness and reproducibility, we standardize each task through multi-stage filtering and augmentation, including resource availability checks, evaluator validation, solution collection, validator construction, and dataset partitioning. Complementing this, iGym provides a unified agent execution environment that supports robust tool use and long-horizon problem solving, ensuring consistent comparisons across diverse systems.

We conduct extensive experiments on InnoGym with several existing agent frameworks. Our findings reveal that current agents still perform significantly below the human state of the art on complex tasks. While some methods demonstrate high novelty, their lack of robustness prevents these innovations from translating into meaningful performance gains. This highlights a key bottleneck in today's agents: in real-world scientific and engineering problems, novelty alone is insufficient—true innovation must combine originality with correctness and effectiveness.

**Our contributions** can be summarized as follows: 1) We propose a principled framework for defining and measuring innovation in AI agents, combining *performance gain* and *novelty* as two complementary evaluation dimensions. 2) We introduce **InnoGym**, the first benchmark specifically targeting innovation potential, consisting of 18 standardized *Improvable Tasks* curated from real-world engineering and scientific domains. 3) We provide **iGym**, a unified agent execution environment that supports reproducible, long-horizon evaluations across diverse systems. 4) We conduct systematic experiments on state-of-the-art agents, uncovering key limitations in robustness and highlighting the gap between novelty and effective performance.

In summary, *InnoGym* establishes both a principled framework and a standardized benchmark for measuring innovation in AI agents, offering a reproducible and cross-domain platform to support future research on systematically evaluating AI's creative and innovative capabilities.

## 2 DEFINING AND MEASURING INNOVATION

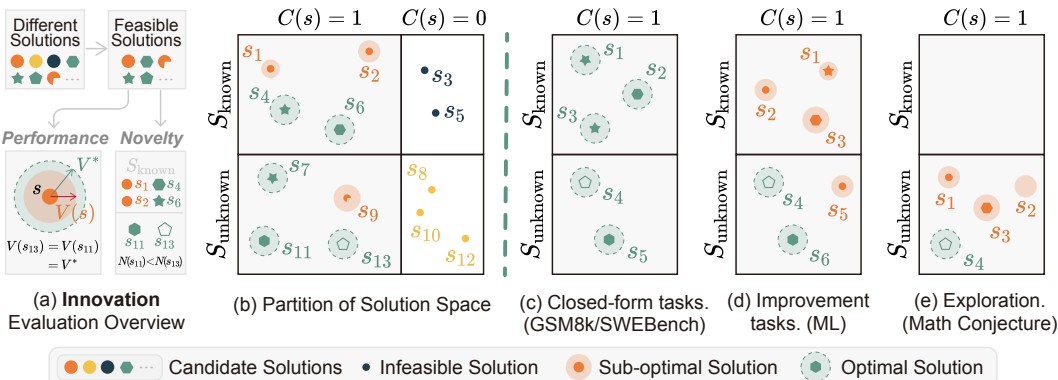

Figure 1: An illustration of our definition framework. **(a)** Core evaluation metrics. Innovation is evaluated along two dimensions: *Performance* ($V$) and *Novelty* ($N$). The colored shapes represent different candidate solutions, while the radius of the background concentric circles corresponds to the magnitude of performance $V(s)$ (larger radius indicates higher performance). **(b)** The solution space is partitioned by feasibility ($C(s)$) and prior knowledge. Feasible solutions (i.e., $C(s) = 1$) are candidates for evaluation. **(c–e)** Categorization of three innovative tasks based on the spatial distribution of solutions relative to the knowledge boundary.

Most existing benchmarks judge agents by answer correctness, overlooking the *solution* that yields the answer. Existing benchmarks for intelligent agents primarily focus on the correctness of the final answer, neglecting the underlying *solution* used to obtain it. Yet intelligence and innovation lie not only in *what* is achieved, but in *how*. Two agents may output the same answer for a problem while

[1]https://www.roadef.org/challenge/

one employs a fundamentally novel solution. This section introduces a framework for quantifying innovation in terms of its *performance* and *novelty*.

## 2.1 TASK AND NOTATION

We define a task as a quadruple $\mathcal{T} = (P, S, V, D)$, where:

- **Problem instance** $P$ contains the task description, constraints, objectives, and evaluation artifacts (e.g., ground-truth answer or test cases). Agents observe $P_{\text{visible}} \subset P$. $P_{\text{hidden}} = P \setminus P_{\text{visible}}$ is for evaluation only.

- **Solution space** $S$ is the set of executable solutions $s \in S$ that can be submitted to solve $P$ (e.g., code, a proof/derivation, an algorithmic strategy).

- **Performance** $V : S \to \mathbb{R}$ quantifies the quality of a solution. We define it as $V(s) = C(s) \cdot R(s)$, where $C(s) \in \{0, 1\}$ checks feasibility or legality (format, execution, constraint satisfaction) and $R(s)$ measures the degree to which a *feasible* solution satisfies the problem's objective (e.g., accuracy, pass rate).

- **Distance** $D : S \times S \to \mathbb{R}_{\geq 0}$ measures dissimilarity between two solutions. A larger value implies greater dissimilarity in the underlying solution. Conceptually, $D$ can be any task-appropriate dissimilarity function (e.g., an embedding-based distance). In our implementation, we instantiate $D$ as an Agent-as-judge score that compares two solutions. See Appendix F for details.

We denote prior-known (human or literature) solutions by $S_{\text{known}} \subseteq S$ and the unknown region by $S_{\text{unknown}} = S \setminus S_{\text{known}}$. For brevity, we omit the task subscript on $(P, S, V, D)$. Then we can define the optimal solution set $S^*$ that achieves the maximal performance score $V^*$:

$$V^* = \max_{s \in S, \, C(s)=1} V(s), \qquad S^* = \{ s \in S \mid C(s) = 1, \, V(s) = V^* \}. \tag{1}$$

However, the optimum $S^*$ is often unknown, intractable, or may not exist for many challenging tasks. Our framework therefore grounds its evaluation in the empirical and dynamic set $S_{\text{known}}$, encompassing the best known solutions ranging from fixed theoretical optima to the evolving SOTA.

## 2.2 DEFINITION AND EVALUATION OF INNOVATION

*What constitutes innovation?* The management theorist *Peter Drucker* famously defined innovation as "*change that creates a new dimension of performance.*" Inspired by this insight, we formalize innovation within our task framework. We define a candidate solution $s$ as innovative if, subject to satisfying feasibility constraints (i.e., $C(s) = 1$), it demonstrates a meaningful differentiation from the set of known solutions $S_{known}$. This differentiation is not one-dimensional; it implies creating value either through superior results or through distinct methodologies. To systematically quantify this, we introduce two complementary metrics: Performance Gain ($G$) and Novelty ($N$).

**Performance Gain ($G$)** measures the performance improvement of a new solution $s$ relative to the frontier of known solutions. We define it as:

$$G(s) = V(s) - V_{\text{known}}^*, \quad V_{\text{known}}^* = \begin{cases} \max_{h \in S_{\text{known}}} V(h), & S_{\text{known}} \neq \emptyset, \\ V_0, & S_{\text{known}} = \emptyset, \end{cases} \tag{2}$$

where $V_0$ is a task-dependent constant baseline for the no-prior case. A positive value of $G$ signifies a super-human performance breakthrough that pushes the state-of-the-art.

**Novelty ($N$)** quantifies dissimilarity to prior solutions and is awarded only to feasible solutions:

$$N(s) = C(s) \cdot \begin{cases} \min_{h \in S_{\text{known}}} D(s, h), & S_{\text{known}} \neq \emptyset, \\ +\infty, & S_{\text{known}} = \emptyset. \end{cases} \tag{3}$$

We only compute novelty for feasible solutions by multiplying by $C(s)$. For a problem with no prior known solutions, any feasible solution is considered maximally novel.

**What kind of solutions are considered innovative?** Given a feasible solution $s$ with performance gain $G(s)$ and novelty $N(s)$, we treat innovation as occupying specific regimes in the $(G, N)$ space. In particular, we regard as **1) breakthrough innovation** those solutions with both high $G$ and high $N$, which substantially improve task value while remaining methodologically distinct from all known baselines. We refer to solutions with high $G$ but relatively low $N$ as **2) performance innovation**: they push the state of the art primarily along the performance axis, often as sophisticated refinements of existing methods. Conversely, solutions with $G(s) \approx 0$ but high $N$ constitute **3) conceptual innovation**: they achieve comparable performance to the best known baseline while introducing a markedly different, feasible paradigm. All other regimes, solutions that are neither better nor different (low $G$, low $N$), or those that are highly novel yet substantially worse than before (large negative $G$ with high $N$), are treated as unsuccessful exploration rather than innovation.

### 2.3 DISCUSSION: A TAXONOMY OF INNOVATIVE TASKS

We categorize task instances according to the spatial distribution of known solutions relative to the feasible region and the knowledge boundary, as illustrated in Fig. 1(c–e). These categories are defined from a human-centered perspective, rather than that of an agent. A formalized definition of each category is given in Appendix D.1.

**Solved Problems:** As shown in Fig. 1(c), tasks with known, optimal solutions, such as problems in MATH (Hendrycks et al., 2021) or SWE-Bench (Jimenez et al., 2024). For these tasks, the performance ceiling is fixed ($V^*_{\text{known}}$ is the optimal score). Innovation is primarily measured by $N(s)$, rewarding new and potentially more efficient methods to reach the known optimal performance.

**Improvable Problems:** As shown in Fig. 1(c), tasks with existing solutions but no known optimum, common in machine learning and optimization challenges. $S_{\text{known}}$ is non-empty but suboptimal. Innovation can be demonstrated either by achieving a new state-of-the-art performance ($G(s) > 0$) or by discovering a fundamentally different method to match current performance (high $N(s)$).

**Exploratory Problems:** As shown in Fig. 1(c), open-ended challenges with no known feasible solutions, such as proving mathematical conjectures or tackling unsolved scientific problems. Here, $S_{\text{known}} = \emptyset$. The first feasible solution ($C(s) = 1$) found by an agent constitutes a monumental innovation, yielding both positive performance gain ($G(s) = V(s) > 0$) and maximal novelty ($N(s) = \infty$). The focus is on the 0-to-1 breakthrough.

## 3 INNOGYM: BENCHMARK AND SYSTEM (IBENCH & IGYM)

**InnoGym** consists of two complementary components: *iBench*, a benchmark designed to evaluate innovation capability, and *iGym*, a unified development and execution environment. iBench covers 18 carefully curated tasks drawn from real-world engineering and theoretical problems. We focus only on *Improvable Tasks*, which leave clear room for improvement in both solution quality and methodology. In contrast, *Solved Problems* (with known optimal solutions) and *Exploratory Problems* (without human baselines or reliable validation) are excluded from the core benchmark, as they either provide no measurable improvement margin or cannot be reliably evaluated.

### 3.1 TASK SOURCES AND TWO-STAGE FILTERING

**Task Sources.** We collect tasks from 2018–2024 across top academic and industrial competitions and workshops (NeurIPS Competitions, KDD Cup, ROADEF, GMCM[2], MLArchSys[3]), as well as from classic NP-hard problems in science and engineering. This produces an initial pool of 197 items, as shown in Fig. 2(a). These tasks span diverse domains and are rooted in real problems that often require multi-disciplinary expertise and sustained collaborative effort, typically ranging from one week to one year. All selected tasks are public, peer-reviewed, and allow the use of a wide range of tools (e.g., CPLEX).

**Stage One: Resource Availability and Affordability.** As shown in Fig. 2(b), we first filter tasks by whether key resources are accessible: datasets, validators or evaluators, leaderboards, and at least

---

[2]Official website: https://cpipc.acge.org.cn/.
[3]Official website: https://sites.google.com/view/mlarchsys

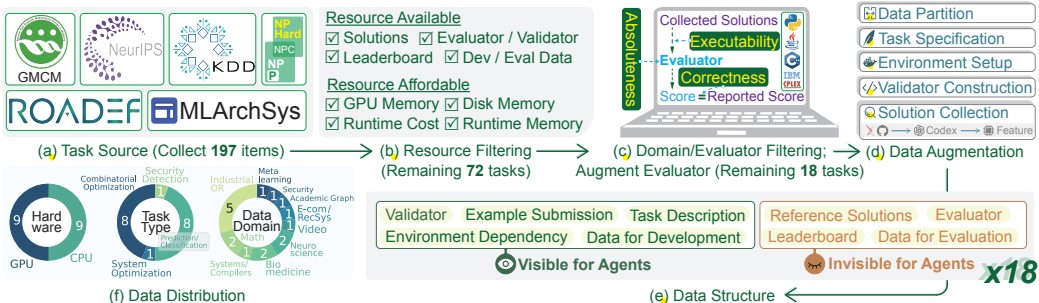

Figure 2: **Dataset curation overview.** We collect 197 tasks from public competitions, filter by resource and evaluator availability, and standardize scoring (executability, correctness, absolute metrics). After augmentation with validators, task specifications, solutions, and environments, the benchmark yields 18 balanced and diverse tasks across domains and hardware.

one reference solution. We also examine computational cost, ensuring that GPU/CPU memory, disk usage, and runtime demands remain feasible. Tasks that pass this stage, and that can be decomposed into multiple sub-tasks, are expanded into individual entries. This stage yields 72 tasks in total.

**Stage Two: Evaluator Quality and Domain Balance.** Next, we validate the correctness and executability of each evaluator. Tasks with unfixable evaluators are removed. To maintain diversity, we further balance across domains, prioritizing newer and more representative tasks. After this process, we obtain 18 high-quality *Improvable Tasks* tasks, as shown in Fig. 2(c)–(e).

### 3.2 ENHANCEMENT AND STANDARDIZATION

To ensure reproducibility and fairness, we augment each task with six types of steps (Fig. 2(d)).

**Task Specification & Environment Setup.** We rewrite descriptions in Markdown, specifying task goals, input/output formats, and submission requirements with clear examples and figures. We package dependencies into reproducible environments (e.g., containerized builds).

**Validator Construction ($C$).** We build or refine validators to check submissions for format, feasibility, and constraints. For example, submitting code tasks validates function signatures, and submitting answer tasks validates fields, ranges, and constraints. See Appx. G.3 for details.

**Solution Collection.** We collect leaderboard solutions and papers. For each solution, we prompt *Codex* (OpenAI, 2025b) with an extraction prompt (see Appx. I.1) to distill its core strategy into a structured representation for novelty analysis. See Appx. G.1 for details.

**Evaluator Normalization ($R$). 1) Absoluteness.** We convert relative or participant-dependent scores (e.g., ROADEF) into instance-level absolute scores, verifying consistency with original rankings (Pearson $\geq 0.9$, Kendall-$\tau \geq 0.8$). **2) Executability.** We ensure evaluators run correctly across languages via standardized command-line or container entry points. **3) Correctness.** We cross-check with public solutions and random baselines, adjusting until leaderboard consistency is achieved. Tasks failing this check are discarded. See Appx. G.2 for details.

**Data Partition.** We split datasets into development (visible) and evaluation (hidden) sets, aligned with leaderboard conventions. All collected resources are explicitly divided into agent-visible and agent-invisible parts, as shown in Fig. 2(e).

### 3.3 TASK FORMALIZATION

Each task instance is formalized as a quadruple $\mathcal{T} = (P, S, V, D)$, consistent with the definitions in Section 2. $P = (P_{\text{visible}}, P_{\text{hidden}})$, where visible parts include descriptions, examples, development

data, and dependencies, while hidden parts include evaluation data, reference solutions $S_{\text{known}}$, and leaderboards. $V(s) = C(s) \cdot R(s)$, where $C$ is the validator (feasibility check) and $R$ is the evaluator (performance measure). $D$ is a distance function used to compute novelty with respect to $S_{\text{known}}$. Agents are given access only to $P_{\text{visible}}$ and $C$, while $P_{\text{hidden}}$, $R$, and $S_{\text{known}}$ remain hidden (Fig. 2(e)).

### 3.4 EVALUATION PIPELINE

The evaluation process proceeds in three stages, as shown in (Fig. 3). **1) Submission.** The agent system produces a solution artifact using only visible data and tools. **2) Performance Evaluation.** The evaluator $R$ computes a score if $C(s) = 1$ (valid submission); otherwise the attempt is rejected. **3) Novelty Evaluation.** The submission is feature-extracted (via Codex prompts, as shown in Appx. H.2) and compared against known solutions $S_{\text{known}}$ using the distance function $D$, yielding a novelty score.

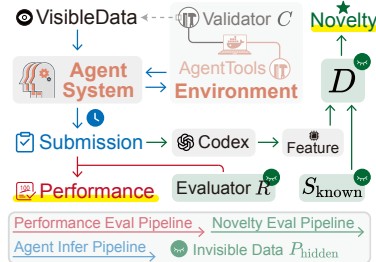

Figure 3: Overview of evaluation pipeline.

Together, these steps provide two complementary measures: performance gain ($G(s)$) and novelty ($N(s)$). Both are required for a task to be considered an innovative success. The key differences between iBench and prior benchmarks are summarized in Table 1.

### 3.5 IGYM: A UNIFIED AGENT EXECUTION ENVIRONMENT

In addition to iBench, our framework also introduces *iGym*, a unified SDK designed to support diverse agent systems and long-horizon problem solving. While existing SDKs such as Open-Hands (Wang et al., 2025), AutoGen (Wu et al., 2023), and LangGraph (LangChain AI, 2024) simplify orchestration, they lack several crucial features needed for our setting, including robust recovery for long-running tasks, native concurrency, and consistent tool management. iGym addresses these limitations by providing a common abstraction layer where agents can interact with environments, tools, and resources under both workflow-style and agent-style paradigms.

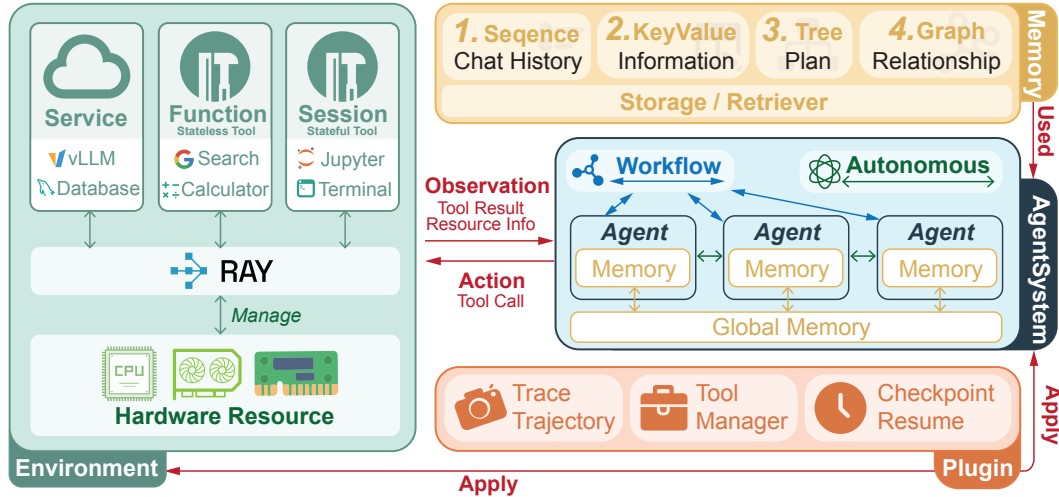

Figure 4: The architecture of iGym.

We show the overview of igym in Fig. 4. Due to space constraints, we defer the full system design, detailed architecture, and illustrative figures to Appx C. There, we present the complete description of iGym, including its asynchronous *Tool Dispatcher*, recovery mechanisms, and examples of concurrent tool usage. This ensures that readers can focus on the benchmark construction in the main paper, while still having access to the complete implementation details of the runtime environment in the appendix.

Table 1: Comparison of existing agent benchmarks and our proposed benchmark. "Ref. Sol." marks whether collected reference solutions are available. "Eval Perf."/"Eval Novelty" denote whether the benchmark explicitly evaluates performance and novelty.

| Benchmark | Source | Data Domain | Ref. Sol. | Difficulty | Compute Profile | Eval Perf. | Eval Novelty. |
|---|---|---|---|---|---|---|---|
| MLAgentBench (Huang et al., 2024) | Kaggle | Machine Learning | ✓ | Easy | GPU | ✓ | ✗ |
| DSBench (Jing et al., 2025) | Kaggle | Machine Learning Science | ✗ | Easy Hard | GPU | ✓ | ✗ |
| MLEBench (Chan et al., 2025) | Kaggle | Machine Learning (Cross-domain) | ✗ | Easy Hard | GPU | ✓ | ✗ |
| ScienceAgentBench (Chen et al., 2025) | Publication | Machine Learning Science | ✓ | Easy Hard | CPU/GPU | ✓ | ✗ |
| MLGym (Nathani et al., 2025) | Kaggle | Machine Learning | ✓ | Easy | GPU | ✓ | ✗ |
| MLRCBench (Zhang et al., 2025b) | NIPS; ECCV; KDD Cup | Machine Learning (Cross-domain) | ✗ | Easy Hard | GPU | ✓ | ✗ |
| InnovatorBench (Wu et al., 2025) | NIPS; ICLR; COLM EMNLP; ACL | Machine Learning | ✓ | Easy Hard | GPU | ✓ | ✗ |
| Ours | NIPS; GMCM; Classical; KDD Cup; ROADEF; MLArchSys; | ML (Cross-domain) Science; OR; Systems; Math | ✓ | Easy Hard | CPU/GPU | ✓ | ✓ |

## 4 EXPERIMENTS

### 4.1 EXPERIMENTAL SETUP

**Models and Agent Scaffolds.** Following MLE-BENCH (Chan et al., 2025), we select three representative agent frameworks, MLAB (Huang et al., 2024), CODEACT (Wang et al., 2025), and AIDE (Jiang et al., 2025) as our main evaluation targets. All agents are executed in the unified iGym environment, so that differences in outcomes primarily reflect the agent design rather than infrastructure. In the main experiments, we use *DeepSeek-v3.1* (DeepSeek-AI et al., 2025b) as the backbone language model. Sec. 4.3 further investigates the performance of *GPT-5* (OpenAI, 2025a) and *Gemini-2.5-Pro* (Gemini Team, 2024) as alternative base models. See Appx. E.1 for details.

**Metrics and Evaluation Protocol.** Our overall evaluation protocol follows Section 3.4. For each solution $s$ submitted by an agent, we measure innovation along the two dimensions defined in Section 2.2: **Performance Gain** $G(s)$ and **Novelty** $N(s)$. We now describe the concrete instantiation of $N(s)$. Novelty is defined as the minimum dissimilarity between $s$ and the known solution space $S_{known}$, where dissimilarity is measured by a distance function $D$. Conceptually, $D$ can be any task-appropriate dissimilarity measure. Here, we instantiate $D$ via an *Agent-as-judge* procedure implemented with *Codex*. For each solution, we first apply an extraction prompt (Appx. I.1) to *Codex* (OpenAI, 2025b) to obtain a structured representation of its core strategy. Given the extracted profiles of an agent solution and a reference solution in $S_{known}$, a novelty-evaluation prompt (Appx. H.2) asks *GPT-5* to rate their methodological dissimilarity along six rubric dimensions, each scored on a 0~4 scale. We average the scores across dimensions, aggregate over all $h \in S_{known}$ via the minimum distance, and then rescale the resulting value to $[0, 100]$ to obtain $N(s)$. See Appx. F.1 for more details. To facilitate comparison across tasks, we further report a **normalized ratio** $Ratio(s) = G(s)/V^*(s)$, where larger values indicate larger relative improvement. A key principle in our evaluation is that novelty is only meaningful when it is effective: high novelty scores are considered important only when accompanied by substantial performance gains. We provide a more detailed analysis of the behavior and reliability of $D$ in Appx. F.

**Implementation Details and Runtime.** Among the 18 tasks in our benchmark, we select 10 tasks as our main evaluation subset. These tasks are relatively more tractable under our computing and engineering constraints (e.g., smaller resource footprint and fewer environment dependencies). For each task–agent–model configuration, we allow up to 12 hours of wall-clock time, or terminate earlier once a submission is completed, whichever comes first. Following MLE-BENCH (Chan

Table 2: Comparison of three agent frameworks with leaderboard upper and lower bounds on the 10 main iBench tasks. "Highest" and "Lowest" are the best and worst known leaderboard scores.

| Competition | LeaderBoard | | MLAB | | | CODEACT | | | AIDE | | |
| --- | --- | --- | --- | --- | --- | --- | --- | --- | --- | --- | --- |
| | Highest | Lowest | Gain | Ratio | Novelty | Gain | Ratio | Novelty | Gain | Ratio | Novelty |
| BEETL(MI) | 76.33 | 31.47 | -35.66 | -0.47 | 66.67 | / | / | / | / | / | / |
| BEETL(Sleep) | 69.23 | 27.91 | -14.64 | -0.21 | 62.50 | / | / | / | -53.62 | -0.77 | 54.17 |
| Belka | 30.62 | 1.26 | -19.02 | -0.62 | 45.83 | -28.14 | -0.92 | 45.83 | -30.01 | -0.98 | 20.83 |
| CirclePacking | 2.635 | 0.96 | -0.43 | -0.16 | 50.00 | -0.008 | -0.003 | 25.00 | -0.25 | -0.09 | 33.33 |
| CDML | 69.90 | 26.50 | / | / | / | / | / | / | / | / | / |
| NPR | 41.21 | 29.53 | -17.10 | -0.42 | 66.67 | -41.16 | -0.99 | 58.33 | / | / | / |
| OAG | 83.45 | 49.95 | -28.59 | -0.34 | 70.83 | -30.38 | -0.36 | 62.50 | -29.87 | -0.36 | 70.83 |
| PTTALC | 48.59 | 14.50 | / | / | / | / | / | / | / | / | / |
| RCIC | 99.76 | 49.15 | / | / | / | -99.67 | -0.99 | 83.33 | -99.67 | -0.99 | 54.17 |
| TrojanDetection | 57.70 | 6.70 | -54.80 | -0.95 | 33.33 | -50.10 | -0.87 | 54.17 | / | / | / |
| Average | 57.94 | 23.79 | -24.32 | **-0.45** | **56.55** | -41.58 | -0.69 | 54.86 | -42.68 | -0.64 | 46.67 |

et al., 2025), we use the same decoding hyperparameters for all agents. Due to computational cost, each configuration is run three times in the main experiments. We report the best score over these three runs, restricted to runs that yield a valid submission. If all three runs for a given configuration fail to produce a valid submission, the corresponding entry is reported as "/" in Table 2.

## 4.2 MAIN RESULTS

Our analysis of the experimental results, presented in Table 2, suggests three primary takeaways regarding the current state of AI agents for innovation. See Appendix E for more details.

**Substantial Performance Gaps on Complex Tasks.** Our primary finding is that existing agents exhibit significant limitations on complex, open-ended problems. Across all evaluated tasks, no agent managed to surpass the state-of-the-art human solutions. On tasks with intricate data formats or complex requirements, such as Cross-Domain-Meta-Learning(CDML) and Perception-Test-Temporal-Action-Localisation-Challenge(PTTALC), all tested agents failed to generate valid and executable solutions. These results highlight a substantial performance gap between current agent capabilities and the robustness required for real-world scientific and engineering problems.

**Differentiation in Existing Frameworks.** Agent frameworks show distinct profiles. MLab leads in both **Performance Gain** and **Novelty**, indicating a rare blend of innovation and execution. Code-Act and AIDE lag on both, likely due to weaker handling of complex file structures and tool use. Notably, CodeAct nears the state of the art on *CirclePacking*, suggesting strength on well-specified mathematical optimization that does not generalize to broader tasks.

**The Primacy of Robustness over Novelty.** Finally, our findings illuminate the intricate relationship between performance and **novelty** across different frameworks. While the three evaluated frameworks exhibited comparable levels of innovation, their performance diverged significantly. This underscores the dominant role of solution correctness and robustness in the context of complex tasks. For example, in RCIC and TrojanDetection tasks, frameworks achieving mid-to-high **novelty** still returned some of the lowest performance scores. This disparity suggests that the primary bottleneck for agents on complex tasks is not a deficit of novel ideas, but rather the inability to translate them into correct and robust implementations. Consequently, ensuring reliable execution quality is the foremost challenge and a critical prerequisite for their real-world applicability.

## 4.3 EXPERIMENTAL ANALYSIS

To further dissect the agent's behavior and the utility of our metrics, we conduct a series of controlled experiments on the challenging Circle Packing problem.

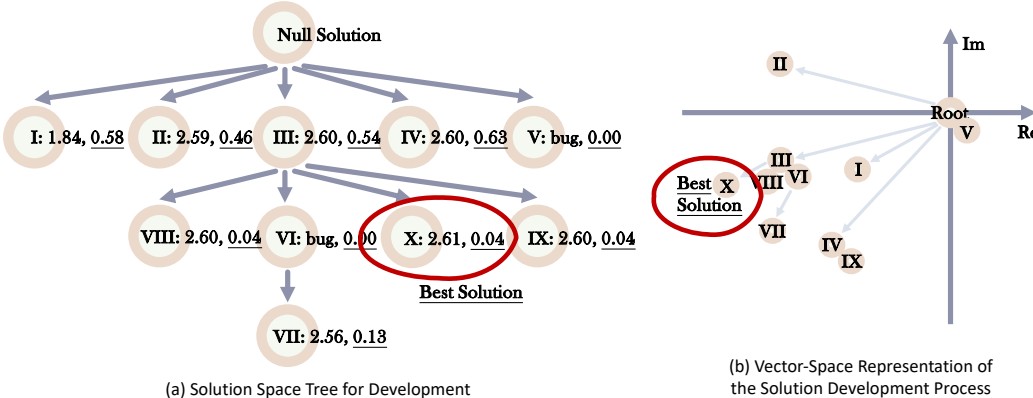

(a) Solution Space Tree for Development

(b) Vector-Space Representation of
the Solution Development Process

Figure 5: An illustration of the solution development process. **(a) Solution Space Tree for Development:** each node represents a candidate solution, where the Roman numeral denotes the iteration order, the first value indicates performance, and the underlined value denotes **novelty**. **(b) Vector-Space Representation of the Solution Development Process:** a complex-plane mapping that jointly encodes **performance gain** (magnitude) and normalized **novelty** (angle), providing a richer interpretation of the development trajectory.

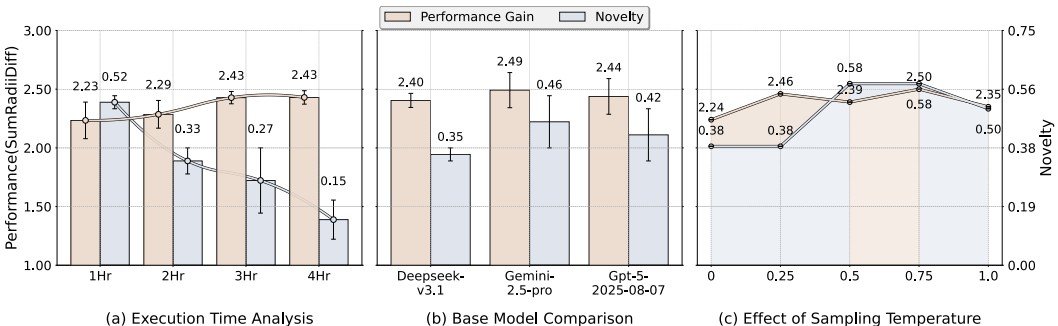

(a) Execution Time Analysis

(b) Base Model Comparison

(c) Effect of Sampling Temperature

Figure 6: Analysis experiments. **(a) Execution Time Analysis:** the effect of varying execution time budgets on **performance gain** and **novelty**, running 3 times. **(b) Base Model Comparison:** the impact of different backbone LLMs, running 3 times. **(c) Effect of Sampling Temperature:** the trade-off different temperature settings.

**The Impact of Prior Knowledge on Innovation.** We first investigate if AIDE can iteratively refine a strong, pre-existing solution. Starting with a solution generated by Gemini-2.5-Pro (sum_radii=2.59, ratio=0.98), we observe that AIDE successfully navigates the solution space to discover superior outcomes. As illustrated in Fig. 5(a), the agent follows an effective trajectory (e.g., 'Null → III → X'), where **Performance Gain** steadily increases. Concurrently, **Novelty** initially peaks—reflecting a significant departure from the starting point—and then gradually decreases as the solution converges toward a local optimum. To better visualize this process, we propose a complex-plane representation. We normalize **Performance Gain** ($G$) to represent the vector's magnitude and the normalized **Novelty** score ($N_{std}$) to define its angle ($2\pi N_{std}$). As shown in (Fig. 5(b)), this mapping reveals directional information obscured by the scalar **novelty** score; solutions with similar N values can represent distinct methodological shifts. This demonstrates that our G and N metrics can be synergistically combined to form a richer, multidimensional representation of the innovation process.

**Temporal Dynamics of Innovation.** Next, we analyze the evolution of $G$ and $N$ over an extended period, where each is measured relative to the previous time step. As shown in Fig. 6(a), $G$ tends to improve over time, while $N$ decreases. This reflects the principle of diminishing returns: as

the solution improves, finding substantial further gains (lower G) becomes harder, and the agent's methodology naturally converges (lower N). Importantly, G remains non-negative throughout, indicating a stable, monotonically improving search process, validating our metrics' ability to capture the typical dynamics of iterative refinement.

**Impact of Foundation Models on Performance.** To isolate the impact of the underlying LLM, we ablate the foundation model while keeping the agent framework constant. The results in Fig. 6(b) show that performance is heavily dependent on the base model's strength. More powerful models like Gemini-2.5-Pro and a hypothetical GPT-5 achieve high scores of 2.49 and 2.44, respectively, closely approaching the AlphaEvolve (Novikov et al., 2025) of 2.65. In contrast, DeepSeek-v3.1 achieves a score of 2.40. This aligns with general community perceptions of these models' capabilities and underscores that agent frameworks act as powerful amplifiers of the base model's intrinsic reasoning and coding abilities, rather than being a substitute for them.

**Exploration-Exploitation Trade-offs at Different Sampling Temperatures.** Finally, we investigate the effect of sampling temperature on agent **performance** and **novelty**. Fig. 6(c) reveals a classic exploration-exploitation trade-off. **Performance Gain** is highest at low temperatures, where the agent exploits known good strategies. Conversely, **Novelty** increases with temperature as the agent is encouraged to explore more diverse, less probable solutions. Our analysis identifies a "sweet spot" in the mid-temperature range (0.5–0.75), where the agent achieves near-optimal performance while significantly boosting methodological **novelty**.

## 5 RELATED WORK

**Evaluation for ML Engineering and Scientific Discovery.** Initial evaluation efforts for LLMs (Jaech et al., 2024; Yang et al., 2025) centered on foundational capabilities in domains like mathematical reasoning (Hendrycks et al., 2021; Cobbe et al., 2021) and code generation (Chen et al., 2021; Jain et al., 2025; Jimenez et al., 2024). The primary metric in these benchmarks is typically correctness, verified through unit tests or exact-match answers. A subsequent line of work evaluates LLMs on open-ended, improvable tasks where the goal is to discover high-performing solutions rather than a single correct one. For instance, MLE-Bench (Chan et al., 2025) challenges agents to develop ML pipelines for Kaggle competitions, directly measuring the solution's value via leaderboard rankings. This focus on performance-driven evaluation is echoed in other ML engineering benchmarks (Huang et al., 2024; Zhang et al., 2025b; Chen et al., 2025; Nathani et al., 2025), as well as in scientific discovery (Majumder et al., 2025) and data science (Zhang et al., 2025a; Jing et al., 2025). The key limitation of this approach is its conflation of solution value with methodological novelty. It fails to distinguish between a genuinely novel method and the effective tuning of a conventional one, as long as both achieve similar performance.

**LLM Agents of Innovation.** Beyond evaluating performance on well-defined tasks, a more ambitious direction assesses the capacity of LLM agents to drive innovation on open-ended scientific problems. Initial efforts in this direction focus on the agent's role as an "idea generator". Existing benchmarks (Ruan et al., 2025; Qiu et al., 2025) and agent systems (Lu et al., 2024; Su et al., 2025; Gottweis et al., 2025) are pivotal in formalizing the assessment of research ideation, but the downstream value of the generated ideas often remains speculative, as they are not executed to solve concrete problems. Building on this demonstrated creative potential, a subsequent line of work has leveraged LLM agents as "problem solvers". These systems translate abstract creativity into concrete breakthroughs, achieving landmark, high-value results on long-standing scientific challenges. AlphaEvolve (Novikov et al., 2025), for example, provided superior solutions for matrix multiplication and the problem of sphere packing.

## 6 CONCLUSION

We introduce **InnoGym**, a benchmark and framework for evaluating the innovation potential of AI agents. By combining performance gain and methodological novelty, InnoGym moves beyond correctness-only evaluation and provides a principled way to measure both effectiveness and creativity. With 18 standardized tasks and a unified execution environment, it enables reproducible, cross-domain comparisons. Experiments reveal that current agents often achieve novelty without robustness, highlighting a persistent gap between creativity and reliable performance.

ETHICS STATEMENT

This study was conducted in full compliance with established ethical standards and research best practices. All data employed are derived and synthesized exclusively from publicly available sources; no proprietary or confidential information was used. Every reference to these data sources is accurately and appropriately cited throughout the paper. We strongly encourage all users of our training dataset to uphold the highest ethical standards, ensuring fairness, transparency, and responsibility in their research. Any use of the dataset that could cause harm or negatively impact society is strictly prohibited.

REPRODUCIBILITY STATEMENT

Due to OpenReview's file size limit, we open-source the benchmark, including the dataset and framework on GitHub at https://github.com/zjunlp/igym. We also detail the experiment settings and prompt in Section 4.1, Appendix E.1, H.2, and I.1.

ACKNOWLEDGEMENT

We would like to express sincere gratitude to the reviewers for their thoughtful and constructive feedback. This work was supported by the National Natural Science Foundation of China (No. 62576307, No. NSFCU23B2055, No. NSFCU19B2027), the Fundamental Research Funds for the Central Universities (226-2023-00138), the 2025 Zhejiang Provincial Center for Disease Control and Prevention Science and Technology Talent Incubation Project (No. 2025-A-04), the 2025 Zhejiang Health Informatics Association Scientific Research Program (Key Project, No. 2025XHZN-Z01), titled "Research on Monitoring and Early Warning Methods of AI Large Model and Infectious Disease Epidemic Data Fusion", undertaken by the Zhejiang Provincial Center for Disease Control and Prevention, Yongjiang Talent Introduction Programme (2021A-156-G), and Information Technology Center and State Key Lab of CAD&CG, Zhejiang University. This work was supported by Ant Group and Zhejiang University - Ant Group Joint Laboratory of Knowledge Graph.

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

## A  USAGE OF LLMs

Throughout the preparation of this manuscript, we used LLMs to assist with improving grammar, clarity, and wording in parts of this work. The use of LLMs was limited to language refinement, with all ideas, analyses, and conclusions solely developed by the authors.

## B  LIMITATIONS

While InnoGym provides a principled framework for evaluating innovation in AI agents, it has several limitations. First, the benchmark currently focuses only on *Improvable Tasks* with clear evaluation pipelines; solved problems and open-ended exploratory problems are excluded, which narrows the scope of applicability. Second, our metrics for performance gain and novelty, though principled, may not capture all dimensions of innovation such as efficiency, interpretability, or long-term impact. Third, novelty is estimated relative to a finite set of known solutions, which may bias

evaluations when prior coverage is limited. Finally, iGym emphasizes reproducibility and robustness but is constrained by computational resources, preventing the inclusion of extremely large-scale tasks. These limitations suggest important directions for extending InnoGym in future work.

## C    iGym

**Motivation.**    Existing SDKs (OpenHands (Wang et al., 2025), AutoGen (Wu et al., 2023), Lang-Graph (LangChain AI, 2024)) simplify agent orchestration and tool use, but cannot rewrite different agent systems under a unified framework, nor do they support long-horizon recovery, resource management, or professional tool integration. iGym addresses these needs with a new SDK that supports diverse system designs, recovery, and concurrency.

**Architecture (Fig. 4).**    iGym consists of two parts:

- **Environment:** A set of tools and resources accessible via a redesigned asynchronous *Tool Dispatcher*, supporting thread-pool or process-pool execution. Agents can launch long-running tasks in parallel with others, monitor progress, and receive real-time results.
- **Agent System:** A collection of agents and memory, interacting with the environment via an *Action → Observation* loop. We support both (1) *workflow mode* (LLM as a function), and (2) *agent mode* (multiple agents with a scheduler, analogous to CPU clock scheduling).

**Key Features.**

- **Recovery:** Workflow mode replays recorded LLM/tool calls; agent mode resumes directly from a saved state.
- **Concurrency:** Native support for parallel tool calls and dependency-aware scheduling.
- **System Compatibility:** A unified abstraction layer allows fair comparison across different agent system designs.

## D    Task

The information for the 18 tasks is shown in Table 3. For each task $\mathcal{T}$, we collect a set of known solutions $S_{\text{known}}(\mathcal{T}) = \{h_1, \ldots, h_m\}$. To characterize how diverse these reference solutions are on each task, we report a diversity statistic *Div*. Let $D_{\text{AGENT}}(\cdot, \cdot)$ denote the method-level distance between two solutions introduced in Section 2 and implemented in Appendix F.1. We define

$$\text{Div}(\mathcal{T}) = \frac{2}{m(m-1)} \sum_{1 \leq i < j \leq m} D_{\text{AGENT}}(h_i, h_j),$$

i.e., the average pairwise distance among all reference solutions for task $\mathcal{T}$ (Rao's quadratic diversity with uniform weights). Because $D_{\text{AGENT}} \in [0, 100]$, the diversity score $\text{Div}(\mathcal{T})$ also lies in $[0, 100]$. Larger values indicate that the known solutions for that task occupy a wider region of method space, making it comparatively harder for new solutions to achieve both high performance and high novelty. When only a single reference solution is available ($m = 1$), $\text{Div}(\mathcal{T})$ is undefined, and we therefore leave the Div. column blank in Table 3.

### D.1    Detail Taxonomy of Innovative Tasks

This appendix gives a more explicit account of the task taxonomy introduced in Section 2, and emphasizes that all categories are defined *per concrete task instance* and *relative to a specific, human-defined goal*.

Consider a task formalized as $\mathcal{T} = (P, S, V, D)$, with problem specification $P$, solution space $S$, value function $V$, and distance function $D$. Let $C : S \to \{0, 1\}$ be the feasibility validator, and let $S_{\text{known}} \subseteq S$ denote the set of known reference solutions (typically human-designed or previously published methods) for this particular instance. With this notation, the three categories in Section 2 can be restated succinctly.

| ID | Task | Title | Source | Number of Ref. Sol. | Div. |
|---|---|---|---|---|---|
| #1 | BEETL(MI) | BEETL 2021 (Motor Imagery) | NeurIPS 2021 Competition | 5 | 66.11 |
| #2 | BEETL(Sleep) | BEETL 2021 (Sleep Staging) | NeurIPS 2021 Competition | 4 | 71.67 |
| #3 | Belka | Predict New Medicines with BELKA | NeurIPS 2024 Competition | 5 | 67.08 |
| #4 | CirclePacking | Circle Packing in a Unit Square | Math / Computational Geometry | 4 | 69.10 |
| #5 | CDML | Cross-Domain Meta-Learning Challenge | NeurIPS 2022 Competition | 3 | 36.80 |
| #6 | NPR | Multilingual Recommendation – Task 1: Next Product | KDD Cup 2023 | 3 | 59.72 |
| #7 | OAG | OAG-Challenge | KDD Cup 2024 | 3 | 70.14 |
| #8 | PTTALC | Perception-Test Temporal Action Localisation (Task 3) | ICCV 2025 | 2 | 25.00 |
| #9 | RCIC | Recursion Cellular Image Classification | NeurIPS 2019 Competition | 3 | 63.20 |
| #10 | TrojanDetection | The Trojan Detection Challenge (LLM Edition) | NeurIPS 2023 Competition | 3 | 54.52 |
| #11 | Roadef2018 | / | ROADEF/EURO 2018 | 1 | / |
| #12 | Roadef2020 | / | ROADEF/EURO 2020 | 1 | / |
| #13 | Roadef2022 | / | ROADEF/EURO 2022 | 1 | / |
| #14 | GMCM 2022 (B Q1) | / | China Graduate Math Modeling 2022 | 2 | 53.21 |
| #15 | GMCM 2022 (B Q2) | / | China Graduate Math Modeling 2022 | 2 | 49.32 |
| #16 | CompilerGym | CompilerGym Benchmark / ISCA 2022 related | ISCA 2022 & Open-source | 3 | 59.63 |
| #17 | 2D Bin Packing | 2D Bin Packing Problem | Classic Combinatorial Optimization | 6 | 43.32 |
| #18 | Graph Coloring | Graph Coloring Problem | Classic Graph Theory | 7 | 57.77 |

Table 3: Tasks in InnoGym, with source benchmark, number of reference solutions $|S_{\text{known}}|$, and diversity score Div. (average pairwise method-level distance under $D_{\text{agent}}$, in ([0,100]))

- **Solved Problem.** A task instance is called *solved* if there exists at least one known solution that is both feasible and optimal with respect to the specified validator and objective. Formally, this means that there exists an $s \in S_{\text{known}}$ such that $C(s) = 1$ and $V(s) = V^*$. Intuitively, the explicit goal encoded by $C$ and $V$ has already been fully achieved by some known solution.

- **Improvable Problem.** A task instance is called *improvable* if at least one known solution passes the validator but none of the known solutions attain the optimal value. Equivalently, there exists an $s \in S_{\text{known}}$ with $C(s) = 1$ and $V(s) < V^*$. In this case, feasibility has been demonstrated, but there is remaining headroom in the explicit objective.

- **Exploratory Problem.** A task instance is called *exploratory* if none of the known solutions pass the validator, that is, if $C(s) = 0$ holds for all $s \in S_{\text{known}}$. Here even feasibility has not yet been established; the immediate challenge is to discover any valid solution at all.

It is important that these categories are *relative to the chosen validator $C$ and value function $V$* rather than to some universal notion of difficulty or optimality. The same underlying real-world problem can fall into different categories if the goal specification changes.

A concrete example is SWE-Bench (Jimenez et al., 2024). Under the standard formulation, the objective is to produce a patch that passes the provided test suite, and the validator checks that the patch applies cleanly and the tests run successfully. Let $s_{\text{known}} \in S_{\text{known}}$ denote the human reference patch. Since $s_{\text{known}}$ passes all tests, it satisfies $C(s_{\text{known}}) = 1$ and achieves the maximal value $V^*$ defined by the test suite. Under this specification, a typical SWE-Bench instance is therefore a *solved* task in the sense above: the stated goal has already been fully met by a known solution.

This does not mean that no better implementation exists in an absolute sense. For example, one could change the objective to "fix the bug with minimal edits while still passing all tests" or to "maximize robustness across an extended test suite". Such a change would alter $V$ and $V^*$, and the same instance might then become *improvable*, since the existing patch could be feasible but suboptimal under the new criterion.

The taxonomy in Section 2 is thus deliberately operational: each task instance is classified as solved, improvable, or exploratory with respect to the explicit, human-defined goal encoded by $(P, C, V)$ and the current set $S_{\text{known}}$, rather than with respect to an abstract, task-agnostic notion of optimality.

## E  ADDITIONAL EXPERIMENT RESULTS

### E.1  SETUP

**Agent scaffolds.** These three scaffolds span complementary design choices for ML engineering agents. MLAB is the *ResearchAgent* from MLAgentBench (Huang et al., 2024), a ReAct-style

| Metric | MLAB | CODEACT | AIDE |
|---|---|---|---|
| $R$ (higher is better) | $-0.62\ [-0.82,\ -0.42]$ | $-0.81\ [-0.98,\ -0.59]$ | $-0.82\ [-0.99,\ -0.60]$ |
| $N$ (higher is better) | $39.58\ [22.08,\ 56.25]$ | $32.92\ [14.58,\ 51.25]$ | $23.33\ [7.50,\ 40.83]$ |

Table 4: Task-level macro-averages and 95% bootstrap confidence intervals for normalized ratio $R$ and novelty $N$ under pessimistic imputation ($R = -1$, $N = 0$ for failure cases). Means are computed over the 10 main iBench tasks, treating tasks as independent observations.

workflow agent that plans in natural language, issues high-level actions such as reading and editing files, executing training scripts, and inspecting logs, and iteratively refines an ML pipeline in a Kaggle-like workspace. CODEACT (Wang et al., 2025) instead unifies all agent actions into executable Python code: the agent generates short programs that directly call libraries, run shell commands, and perform self-debugging through repeated code execution, which has been shown to significantly improve success rates across tool-using benchmarks and underlies the CodeActAgent implementation used in MLE-Bench (Chan et al., 2025). Finally, AIDE (Jiang et al., 2025) is a tree-search–based ML engineering agent that views the task as code optimization: starting from an initial solution, it repeatedly proposes code edits, runs training and evaluation, and branches on promising variants, reusing and refining strong configurations to trade compute for performance and achieve state-of-the-art results on Kaggle competitions. Together, MLAB, CODEACT, and AIDE cover workflow-style planning, code-centric action spaces, and search-based exploration, providing a diverse set of agent scaffolds for evaluating innovation on InnoGym.

### E.2 STATISTICAL ANALYSIS FOR TABLE 2

In Table 2, a "/" entry indicates that all runs for a given task, agent failed to produce a valid submission within the budget. The main analysis computes macro-averages over tasks on which each agent has at least one valid solution. To explicitly incorporate failure cases ("/" entries) and to perform hypothesis tests on our key metrics, we adopt the following *pessimistic* imputation scheme.

For any task, an agent with no valid submission, we assign the minimum possible normalized ratio $R = -1$ and a novelty score $N = 0$. Intuitively, an agent that never returns a valid solution to a task contributes neither usable performance nor innovative methodology to that task. Under this encoding, each agent framework obtains a ratio and novelty score on every one of the 10 main tasks, and we treat tasks as independent units for statistical analysis.

**Macro-averages and confidence intervals.** For each framework $f$ and metric $M \in \{R, N\}$, we compute the task-level macro-average

$$\hat{\mu}_f^M = \frac{1}{T} \sum_{t=1}^{T} M_{f,t},$$

and estimate uncertainty via non-parametric bootstrap over tasks ($B = 10{,}000$ resamples). The 95% confidence intervals are obtained from the empirical 2.5th and 97.5th percentiles of the bootstrap distribution of $\hat{\mu}_f^M$. The resulting averages and confidence intervals are reported in Table 4.

Even under this pessimistic treatment of failure cases, all three agent frameworks remain far from the best-known solutions on average (ratios close to $-1$), with MLAB attaining the best normalized ratio and the highest average novelty.

**Paired tests across frameworks.** To compare frameworks on $R$ and $N$, we treat tasks as paired observations. For each metric $M \in \{R, N\}$ and framework pair $(f, f')$, we define the task-wise difference $d_t^M = M_{f,t} - M_{f',t}$ and estimate the mean difference

$$\Delta_{f,f'}^M = \frac{1}{T} \sum_{t=1}^{T} d_t^M$$

together with a 95% confidence interval and an approximate two-sided bootstrap $p$-value. Specifically, we perform non-parametric bootstrap resampling over tasks ($B = 10{,}000$ resamples) and

| Metric | Pair | $\Delta$ | 95% CI | $p_{\text{boot}}$ |
|---|---|---|---|---|
| $R$ | MLAB & CODEACT | +0.20 | [0.01, 0.40] | 0.035 |
| $R$ | MLAB & AIDE | +0.20 | [0.05, 0.37] | 0.007 |
| $R$ | CODEACT & AIDE | +0.01 | [−0.06, 0.05] | 0.785 |
| $N$ | MLAB & CODEACT | +6.67 | [−18.75, 30.42] | 0.575 |
| $N$ | MLAB & AIDE | +16.25 | [−5.00, 37.08] | 0.133 |
| $N$ | CODEACT & AIDE | +9.58 | [−10.00, 29.17] | 0.333 |

Table 5: Paired bootstrap tests over tasks under pessimistic imputation ($R = -1$, $N = 0$ for failures). $\Delta$ denotes the macro-average difference between the first and second framework ($f - f'$). We report 95% bootstrap confidence intervals and approximate two-sided bootstrap $p$-values over the 10 main tasks.

compute $\Delta^M_{f,f'}$ on each resample; the 2.5th/97.5th percentiles of this bootstrap distribution give the confidence interval, and the bootstrap $p$-value is obtained as twice the smaller of the fractions of bootstrap means above and below zero. Results are summarized in Table 5.

Under this pessimistic encoding, MLAB consistently achieves higher macro-average ratio $R$ than both CODEACT and AIDE; the improvement is statistically significant in both comparisons according to the bootstrap test. For novelty $N$, MLAB also has the highest mean, but differences between frameworks are not statistically significant at the 0.05 level given the small number of tasks. We therefore interpret the $N$ results as descriptive trends, and use $R$ as the primary performance metric for formal cross-framework comparisons.

## E.3 OTHER ANALYSIS

| Task | MLAB | CODEACT | AIDE |
|---|---|---|---|
| BEETL (MI) | 100% | 0.00% | 0.00% |
| BEETL (Sleep) | 100% | 0.00% | 100% |
| Belka | 100% | 100% | 100% |
| CirclePacking | 100% | 100% | 100% |
| CDML | 0.00% | 0.00% | 0.00% |
| NPR | 33.33% | 33.33% | 0.00% |
| OAG | 100% | 100% | 66.66% |
| PTTALC | 0.00% | 0.00% | 0.00% |
| RCIC | 0.00% | 33.33% | 33.33% |
| TrojanDetection | 33.33% | 33.33% | 0.00% |

Table 6: Submission success rates of different agents across 10 tasks over 3 runs.

**Submission success rates.** We report the submission success rates across three runs in Table 6. Notably, in 2 out of the 10 sampled tasks (CDML and PTTALC), no agent was able to produce a valid submission. Combined with the performance gaps shown in Table 2, *these results confirm that our benchmark is distinctly future-oriented.* Unlike prior tasks that are often solved in minutes, these high-value challenges—derived from real-world scientific and engineering competitions—require continuous iteration and runtimes spanning tens of hours. We believe these are precisely the complex, long-horizon problems that the next generation of LLMs and agents must master, and we invite the community to tackle these rigorous standards to drive the next leap in machine intelligence.

**Encourage Novelty Explicitly.** We further investigated the effect of explicitly prompting (see Fig. 7) the AIDE agent to prioritize innovation on three tasks, with results reported in Table 7. While this strategy successfully improved the Novelty scores, significantly in the case of CirclePacking, it consistently resulted in a decline in Performance Gain. This demonstrates that exploratory behavior imposes a cost on agent performance. Consequently, we conclude that the pursuit of methodological novelty must not come at the expense of solution correctness, and future agents must learn to balance creativity with effectiveness.

| Task | Gain (Ratio) | | Novelty | |
|------|-----------|------------|----------|------------|
| | Baseline | Innovative | Baseline | Innovative |
| BETTL(sleep) | **-53.62 (-0.77)** | -54.56 (-0.79) | 54.17 | **58.33** |
| OAG | **-29.87 (-0.36)** | -31.76 (-0.38) | **70.83** | 50.00 |
| CirclePacking | **-0.25 (-0.09)** | -0.37 (-0.14) | 35.33 | **58.33** |

Table 7: Performance comparison of agent AIDE (Jiang et al., 2025) behavior with and without innovation prompts across three tasks. Baseline refers to standard prompting, while Innovative indicates prompts explicitly encouraging creative solutions.

## F  VALIDATING THE DISTANCE FUNCTION $D$ AND NOVELTY METRIC $N$

In the main text, we formalize each task as $\mathcal{T} = (P, S, V, D)$, where $D(s_1, s_2)$ measures the distance between two solutions $s_1$ and $s_2$. The novelty of a candidate solution $s$ is then defined in terms of its distance to the set of known solutions $S_{\text{known}}$ (Eq. 3). Intuitively, a solution is more novel if, in terms of "how it solves the problem," it is far from previously observed solutions.

Here, we instantiate $D$ with an Agent-based pipeline, which we denote by $D_{\text{AGENT}}$. The pipeline has two stages: (i) an *extraction* step that summarizes each solution into a standard representation, and (ii) a *comparison* step that scores method-level differences along a small set of dimensions. Both stages are implemented by prompting an agent named Codex (OpenAI, 2025b). In this section, we first recap the pipeline, then describe a triplet-based protocol to validate $D_{\text{AGENT}}$ against human judgments, and finally present two experiments: one on code-level equivalents (EquiBench (Wei et al., 2025a)), and one on human-collected method triplets across different AI subfields.

### F.1  RECAP: NOVELTY EVALUATION PIPELINE

**Extraction.**  For each solution $s$ (including both historical leaderboard entries and new agent submissions), we first run an *extraction* prompt (Appx. I.1) over the entire solution repository. The codex agent produces two standardized artifacts:

- **summary.md**: A structured Markdown file clearly describing the solution's core ideas, data processing pipeline, and model architecture in natural language.
- **pseudocode.tex**: A LaTeX-formatted pseudocode file, outlining the solution's logic and key steps in an algorithmic format.

These two files serve as a normalized, human-readable representation of the solution, stripping away incidental details such as file layout or naming conventions.

**Comparison.**  Given two solutions $s_1$ and $s_2$, along with their corresponding summary.md and pseudocode.tex, we then run a second *comparison* prompt (Appx. H.2). The prompt asks the GPT-5 to act as a reviewer and assess how different the two solutions are along a fixed set of method dimensions $\mathcal{K}$. For each dimension $k \in \mathcal{K}$, the agent assigns a discrete score

$$d_k(s_1, s_2) \in \{0, 1, 2, 3, 4\},$$

where 0 means "essentially the same" and 4 means "completely different paradigm" for that particular aspect. We then aggregate these per-dimension scores into a single distance. First, we normalize each $d_k$ to the range $[0, 1]$ by dividing by 4, then we average over all dimensions, and finally rescale to a 0–100 scale:

$$D_{\text{AGENT}}(s_1, s_2) = \frac{1}{|\mathcal{K}|} \sum_{k \in \mathcal{K}} \frac{d_k(s_1, s_2)}{4} \times 100. \tag{4}$$

In all experiments in the main paper, we use $D_{\text{AGENT}}$ as the concrete instantiation of $D$ when computing novelty scores.

### F.2  TRIPLET-BASED VALIDATION PROTOCOL

Having defined $D_{\text{AGENT}}$, we now ask whether it agrees with human intuition about method similarity and novelty. To answer this, we design a simple triplet-based protocol.

| | $D_{\text{AGENT}}(A, B)$ | $D_{\text{AGENT}}(A, C)$ |
|---|---|---|
| Mean over 50 triplets | 1.00 | 9.75 |

Table 8: EquiBench results over 50 triplets. Mean $D_{\text{AGENT}}$ scores on a 0–100 scale.

| ID | $D_{\text{AGENT}}(A, C)$ | $D_{\text{AGENT}}(A, B)$ | Human(A,C) | Human(A,B) |
|---|---|---|---|---|
| Case 1 | 29.17 | 4.17 | 33.33 | 0.00 |
| Case 2 | 29.17 | 0.00 | 33.33 | 0.00 |
| Case 3 | 0.00 | 4.17 | 12.50 | 0.00 |
| Case 4 | 54.17 | 0.00 | 29.16 | 0.00 |
| Case 5 | 0.00 | 0.00 | 0.00 | 0.00 |
| Case 6 | 0.00 | 0.00 | 0.00 | 0.00 |
| Case 7 | 0.00 | 0.00 | 8.33 | 0.00 |
| Case 8 | 12.50 | 0.00 | 16.66 | 0.00 |
| Average | 15.63 | 1.04 | 16.66 | 0.00 |

Table 9: Comparison of $D_{\text{AGENT}}$ scores against human judgments on the 8 sub-sampled EquiBench triplets. Both agent and humans score the distance from a base solution ($A$) to an algorithmic variant ($C$) and a superficial variant ($B$). Scores are 0–100. See Table 10 for the correlation scores between human and agent judgments.

Each evaluation instance is a triplet $(A, B, C)$, where $A$ is a *base solution* (reference method for a given task); $B$ and $C$ are two alternative solutions to the same task. In our settings, one can think of $A$ as an existing known solution, and $B$ and $C$ as two new solutions proposed by different agents. When we construct triplets for validation, we explicitly choose $B$ and $C$ so that, within each triplet, **we have a clear expectation $B$ is relatively closer to $A$ in terms of method; $C$ is *relatively farther* from $A$, and therefore more novel.** This gives us a ground-truth notion of "who should be more novel relative to $A$" for each triplet.

For each triplet, we compute $D_{\text{AGENT}}(A, B)$ and $D_{\text{AGENT}}(A, C)$. We also collect human judgments: annotators rate how different $B$ and $C$ are from $A$ on a 0–100 scale based standard shown in Appx. H.2, producing *Human*$(A, B)$ and *Human*$(A, C)$. Higher scores mean that the solution is viewed as more methodologically distant and therefore more novel.

We compare the agent against humans in two ways. **First**, we examine score-level correlation, computing Pearson and Spearman correlations between agent and human scores, separately for the $(A, B)$ and $(A, C)$ pairs. **Second**, we look at triplet-level agreement: for a given triplet, we say the agent and the human agree if they both judge $B$ to be more novel, both judge $C$ to be more novel, or both judge them to be roughly tied. The agreement rate is the fraction of triplets where this holds.

We apply this protocol to two datasets: code-level equivalents from EquiBench (Wei et al., 2025a), and human-constructed method triplets from three AI subfields.

## F.3 EXPERIMENT 1: EQUIBENCH CODE-LEVEL SANITY CHECK

**Data Collection.** EquiBench (Wei et al., 2025a) groups functionally equivalent programs into several categories. We focus on two:

- `OJ_A`: functionally equivalent solutions that use different algorithms or implementations;
- `OJ_V`: purely superficial variants with identical logic, such as variable renaming.

This gives us a straightforward sanity check: a reasonable method distance should assign a non-trivial distance to `OJ_A` pairs, but a near-zero distance to `OJ_V` pairs.

We randomly sample 50 triplets of the form $A = $ base solution, $B = $ `OJ_V` variant, $C = $ `OJ_A` variant, and compute $D_{\text{AGENT}}(A, B)$ and $D_{\text{AGENT}}(A, C)$ for each. Table 8 reports the average agent scores over all 50 triplets.

| Dataset | # Triplets | Pearson $r$ (A,B) | Pearson $r$ (A,C) | Spearman $\rho$ (A,B) | Spearman $\rho$ (A,C) | Triplet agreement |
|---|---|---|---|---|---|---|
| EquiBench (annot.) | 8 | *n/a* | 0.84 | *n/a* | 0.87 | 6/8 = 75% |
| Human triplets | 3 | 1.00 | 0.99 | 1.00 | 1.00 | 3/3 = 100% |

Table 10: Correlation between $D_{\text{AGENT}}$ and human judgments. For EquiBench A–C pairs, all human scores are zero, so correlations are not defined.

| Domain | Reference $A$ | Within-paradigm $B$ | Cross-paradigm $C$ |
|---|---|---|---|
| Self-supervised vision | SimCLR (Chen et al., 2020) | MoCo (He et al., 2020) | BYOL (Grill et al., 2020) |
| Neural radiance fields | NeRF (Mildenhall et al., 2020) | NSVF (Liu et al., 2020) | Instant-NGP (Müller et al., 2022) |
| Long-context modeling | Glyph (Cheng et al., 2025) | DeepSeek-OCR (Wei et al., 2025b) | LongRoPE (Ding et al., 2024) |

Table 11: Human-collected method triplets across three domains. In each triplet, $A$ and $B$ are concurrent methods within the same paradigm, while $C$ addresses the same problem with a qualitatively different modeling approach.

For human evaluation, we sub-sample 8 of the 50 triplets and ask three Computer Science graduate students with strong programming backgrounds to score each $(A, B)$ and $(A, C)$ pair on the 0–100 scale, based on standard at Appx. H.2. We average their scores per pair. Table 9 shows the detailed numbers. The alignment between human and agent judgments, quantified by correlation, is detailed in Table 10.

**Results and takeaway.** Over 50 triplets, `OJ_A` variants are on average substantially farther from the base solution than `OJ_V` variants (9.75 vs. 1.00), which is exactly what we want. On the 8 human-annotated triplets, the agent and humans give very similar mean scores: around 16 for $(A, C)$, and essentially 0 for $(A, B)$. Correlations are high for the $A, C$ pairs, and the agent matches human preferences (which of $B$ or $C$ is more novel) on 6 out of 8 triplets. **In short, $D_{\text{AGENT}}$ ignores superficial edits and reacts to real algorithmic changes in much the same way as human programmers.**

### F.4 EXPERIMENT 2: HUMAN-COLLECTED METHOD TRIPLETS

**Data Collection.** We next look at higher-level methodological differences. We construct three triplets from three AI subfields. In each triplet, $A$ is a reference method, $B$ is a within-paradigm method, and $C$ is a cross-paradigm method that is widely viewed as more novel. We summarize the collected triplets in Table 11.

For each domain, we recruit one PhD student working in that subfield (and with reviewing experience) to rate $(A, B)$ and $(A, C)$ on the same 0–100 scale as before based on Appx. H.2. We also compute $D_{\text{AGENT}}(A, B)$ and $D_{\text{AGENT}}(A, C)$ using the same pipeline. Table 12 summarizes the scores. The alignment between human and agent judgments, quantified by correlation, is detailed in Table 10.

**Results and takeaway.** Across the three domains, both the agent and the human experts consistently judge $C$ to be more novel than $B$ relative to $A$, and the average scores are close in magnitude. Despite the tiny sample size, the score-level correlations are essentially perfect as shown in Table 10, and the triplet-level agreement rate is 3/3. This suggests that $D_{\text{AGENT}}$ is sensitive not only to code-level changes, but also to the kind of paradigm shifts that researchers care about.

### F.5 OVERALL SUMMARY

Putting the two experiments together, we see a consistent picture. On EquiBench, $D_{\text{AGENT}}$ treats purely cosmetic variants as essentially zero-distance while assigning noticeably larger distances to algorithmically different solutions, and its preferences align well with those of human programmers. On cross-domain method triplets, it agrees with domain experts on which methods are more novel

| Triplet (Domain) | $D_{\text{AGENT}}(A, B)$ | $D_{\text{AGENT}}(A, C)$ | Human$(A,B)$ | Human $(A,C)$ |
|---|---|---|---|---|
| Self-supervised vision | 29.16 | 45.80 | 33.33 | 54.17 |
| Neural radiance fields | 50.00 | 54.17 | 41.67 | 66.67 |
| Long-context modeling | 66.00 | 95.83 | 50.00 | 100.00 |
| Average | 48.39 | 65.27 | 41.67 | 73.61 |

Table 12: Human-collected method triplets. Each row corresponds to one subfield; scores are on a 0–100 scale. See Table 10 for the correlation scores between human and agent judgments.

and produces scores on roughly the same scale. In practice, this means that $D_{\text{AGENT}}$ is doing what we want: it measures method-level differences rather than surface edits, and its notion of novelty tracks human intuition both at the code level and at the level of high-level method design.

## G  DETAILS OF BENCHMARK CONSTRUCTION

### G.1  SOLUTION COLLECTION

To construct a relatively comprehensive and high-quality set of known solutions ($S_{\text{known}}$) for the 18 tasks in *iBench*, we executed a systematic, multi-stage collection, filtering, and post-processing pipeline that ensures the solutions used to calculate Novelty are robust and representative.

**Sources of Candidate Solutions.**  Our collection strategy was primarily divided into two categories based on task type:

- **Classical NP-hard Problems:** For classical combinatorial optimization or mathematical problems (e.g., `#17` 2D Bin Packing and `#18` Graph Coloring in Table 3), we first consulted authoritative operations research and algorithm design textbooks, as well as academic surveys. This allowed us to identify standard methods, classical heuristics (e.g., greedy algorithms, simulated annealing), and exact algorithms in the field.

  Subsequently, we searched for prominent open-source implementations of these classical methods, particularly those widely cited in academia or implemented in standard libraries, and added them to our candidate set.

- **Specific Competition Tasks:** For tasks originating from academic or industry competitions such as `NeurIPS`, `KDD Cup`, and `ROADEF`, we collected solutions through three main channels:

  (i) **Academic Literature:** For tasks from academic venues (e.g., NeurIPS Competitions), we used tools like `Google Scholar` to find methodology papers that cited the original competition-organizing paper. We prioritized papers that detailed their methodology and provided public source code.

  (ii) **Public Leaderboards and Code Repositories:** We meticulously reviewed the official leaderboards for each competition. We focused on collecting high-ranking entries where the authors publicly shared their full solutions (e.g., via `GitHub repositories` or `Kaggle Notebooks`).

  (iii) **LLM-Assisted Search:** During the search process, we utilized `ChatGPT` to assist in generating diverse search keywords (e.g., alternative task names, related algorithm families) and to help quickly summarize technical blogs and forum posts to discover additional potential candidates.

**Validation and Filtering Process.**  To ensure the comprehensiveness and quality of the $S_{\text{known}}$ set, we employed a rigorous validation process:

1. **Independent Search and Merging:** For each task, we assigned three team members to independently conduct the collection process described above. This cross-validation approach was designed to maximize coverage of solutions from different sources and ensure the comprehensiveness of the candidate set.

2. **Reproducibility Filtering:** All collected candidate solutions were tested within the standardized environment provided by iGym. We executed all relevant solutions and **retained only** those that met both of the following conditions:

   (a) **Executability:** The code had to compile and run successfully without substantial modification.

   (b) **Performance Consistency:** The reproduced performance score had to meet or closely approach the level reported in the original paper or on the leaderboard.

3. **Final Set:** Through this strict filtering process, we removed entries that were inoperable or whose performance was not reproducible. The final number of available, validated solutions for each task is presented in Table 3.

**Structural Extraction of Solutions.** To perform systematic novelty analysis on the filtered $S_{\text{known}}$ set, we needed to abstract each solution from its code *implementation* to its core *methodology*.

As described in Section 3.2 of the main paper, for each retained solution, we used *Codex* with a carefully designed Extraction Prompt (see Appendix I.1) to automatically "distill" its core strategy.

This process generated two standardized output files for each solution in $S_{\text{known}}$:

- **summary.md:** A structured Markdown file clearly describing the solution's core ideas, data processing pipeline, and model architecture in natural language.

- **pseudocode.tex:** A LATEX-formatted pseudocode file, outlining the solution's logic and key steps in an algorithmic format.

These two structured representations collectively form the baseline database used for Novelty Evaluation.

## G.2 EVALUATOR NORMALIZATION

To ensure fair, consistent, and reliable benchmarking across 18 diverse tasks, we implemented a rigorous three-part normalization process. This process guarantees that every evaluator adheres to our standards of **Absoluteness**, **Executability**, and **Correctness**.

### G.2.1 ABSOLUTENESS

**Rationale:** To meaningfully quantify Performance Gain ($G$, see equation 2), our framework requires an absolute scoring metric. Relative scores, such as rankings, are insufficient as they cannot measure the magnitude of an agent's improvement over the state-of-the-art.

**Problem:** Several of our collected tasks, most notably the three ROADEF challenges (Roadef2018, Roadef2020, Roadef2022), used rank-based scoring on their official leaderboards.

**Solution:** We developed a procedure to convert these rank-based systems into absolute scales.

1. We first collected all available scores from the public leaderboards to identify the best-known (highest) and worst-known (lowest) performance scores.

2. These maximum and minimum values were then fixed as static hyperparameters for our new evaluation function.

3. We applied a **logarithmic normalization function** to transform the raw scores onto a consistent, absolute scale. This allows any new, valid solution to receive an absolute score, making it directly comparable to existing solutions.

**Validation:** To ensure our new absolute metric preserved the qualitative integrity of the original leaderboards, we validated its consistency with the original rankings. We calculated the correlation between our normalized scores and the original ranks using Spearman's rank correlation ($\rho$) and Kendall's rank correlation ($\tau_a$). As shown in Table 13, the high correlation values confirm that our absolute scores strongly maintain the original relative ordering of solutions.

Table 13: Correlation between our normalized absolute scores and the original ROADEF leaderboard rankings.

| Task | Spearman's $\rho$ | Kendall's $\tau_a$ |
|------|------------------|-------------------|
| Roadef2018 | 0.960 | 0.877 |
| Roadef2020 | 0.960 | 0.859 |
| Roadef2022 | 0.982 | 0.924 |

### G.2.2 EXECUTABILITY

**Rationale:** Our iGym environment requires a unified interface to trigger any task's evaluation from a Python-based workflow, regardless of the evaluator's original implementation.

**Problem:** The evaluators we collected were highly heterogeneous. Some were only described in documentation (requiring us to implement them), while others were provided as binaries or source code in different languages (`Java`, `C`, `C++`). Many of these had strict and often conflicting environment dependencies.

**Solution:** We standardized all evaluators by containerizing the non-Python components.

1. For every evaluator that was not a simple Python script, we built a Docker container. This container encapsulated all necessary dependencies, such as a specific Java Runtime Environment or C compiler.
2. We then created lightweight Python wrappers that use a subprocess to call the executable within the container.

This abstraction allows the iGym framework to treat every evaluator identically: as a simple Python function call that takes a submission file and configuration as input, and returns a score.

### G.2.3 CORRECTNESS

**Rationale:** The evaluator's correctness is paramount. A faulty evaluator could reward invalid solutions or penalize valid ones, rendering the benchmark useless.

**Problem:** We needed to verify the correctness of all 18 evaluators, especially those we implemented ourselves from descriptions.

**Solution:** We employed a multi-pronged validation strategy:

- **Known Solution Verification:** For tasks with known solutions (i.e., $S_{known}$) and reported scores, we executed these solutions in our environment. We verified that our evaluator produced a score that was identical or (in the case of stochastic algorithms) statistically very close to the one reported on the official leaderboard.
- **Baseline Sanity Checks:** For non-code submission tasks (e.g., classification), we generated trivial or random submissions. For example, we would create a submission file that predicted the same label for every instance. We then verified that this submission produced a valid and appropriately lower score.
- **Monotonicity Check:** We compared the scores from the baseline submissions against the scores from known high-performing solutions. This was a simple but critical check to ensure that our evaluators correctly scored better solutions higher than trivial ones.

If an evaluator failed any of these checks, it was flagged for review. We iteratively debugged and refined the evaluator's logic (or its containerized environment) until it successfully passed all correctness tests.

### G.3 VALIDATOR CONSTRUCTION

For each task, we implement a dedicated validator to process the agent's submission before it is passed to the evaluator. The primary purpose of this validator is to filter out ill-formed or invalid

submissions—a failure mode frequently encountered in practice. We support two types of submissions:

1. **Code submissions.** In this mode, the agent submits a code file that is required to implement a prescribed interface. The validator first checks that the expected entrypoint function is present and that its inputs and outputs conform to the task specification (for example, the argument list and return type). It then executes the function on a small, fixed test input to verify that the code runs without errors and produces a result of the correct type. If any of these checks fail, such as a missing entry point, a runtime exception, or a mismatched return type, the submission is rejected and the validator returns an error instead of a score.

2. **Answer-file submissions.** In this mode, the agent submits a structured output file, typically in `CSV` or `JSON` format. The validator enforces that the file type matches the required format and that its schema (for example, column names and field structure) agrees with the task specification. It also checks basic constraints on individual fields, such as required presence, allowed value ranges, or discrete label sets. Submissions that violate any of these structural or value-level constraints are rejected.

All validators are implemented as purely procedural code, invoking no language models or other stochastic components. This design renders the validation process fully deterministic: a given submission will always produce the same validation outcome. This simplifies debugging and ensures that evaluation results are precisely reproducible.

## H    PROMPT

We designed two structured prompts to systematically analyze and compare solutions: one for methodology extraction and one for solution comparison. Their purposes are briefly described below.

### H.1    EXTRACTION PROMPT

This prompt instructs the model to act as a senior ML engineer and technical writer, reading the entire code repository (and any accompanying paper) to reconstruct the high-level solution methodology. It focuses on problem context, modeling choices, data flow, training strategy, and evaluation protocol, while omitting low-level implementation details. The output is a structured markdown summary (summary.md) and a LaTeX-style pseudocode (pseudocode.tex), providing a consistent and human-readable description of the solution for downstream analysis.

### H.2    COMPARISON PROMPT

This prompt evaluates the similarity between two solutions—an Agent Solution and a Baseline—across six dimensions (e.g., problem framing, methodology, architecture, experiment design). For each dimension, the model assigns a score from 0 (completely similar) to 4 (completely different) with a brief justification. The resulting JSON object enables quantitative, reproducible comparison of solution approaches.

## I    DISCUSSION

### I.1    PROPERTIES OF INNOVATION

We posit that innovation exhibits *Contextual Relativity* and *Temporal Dynamicity*.

**Contextual Relativity.**    Innovation is not an absolute scalar but a relative metric contingent on the specific task. **First**, the innovation of a solution $s$ is strictly defined with respect to the task formulation $\mathcal{T}$. A solution may be innovative for task $\mathcal{T}_A$ yet trivial for $\mathcal{T}_B$. This distinction is particularly relevant when tasks share the same problem space $P$ but differ in their value functions $V$. For instance, a solution that maintains parity in accuracy but significantly reduces resource consumption becomes innovative only if the task definition $\mathcal{T}_B$ explicitly incorporates computational cost into

$V$, whereas it might be deemed trivial under a formulation $\mathcal{T}_A$ that values accuracy alone. **Second**, innovation is measured relative to the reference set of known solutions, $\mathcal{S}_{\text{known}}$. Since different tasks have distinct baselines, a comprehensive $\mathcal{S}_{\text{known}}$ is essential for precisely estimating the novelty score $N(s)$ and performance gain $G(s)$, thereby quantifying how significantly a candidate deviates from the status quo. **Finally**, the threshold for what constitutes a "non-trivial" improvement is also relative. In mature domains—such as the Circle Packing problem, where optimization has converged near the theoretical limit, even marginal performance gains or minor methodological refinements are sufficient to constitute valid innovation.

**Temporal Dynamicity.** Innovation is inherently time-variant; a method considered standard practice today was likely novel at its inception. We model this evolution by explicitly updating the $S_{\text{known}}$ over time: $S_{\text{known}}^{(t)} \rightarrow S_{\text{known}}^{(t+1)}$. Once a solution $s$ is deemed innovative at time $t$ and accepted, it is assimilated into the baseline at $t+1$ (i.e., $\mathcal{S}_{\text{known}}^{(t+1)} \leftarrow \mathcal{S}_{\text{known}}^{(t)} \cup \{s\}$). This monotonic update implies that the measured novelty of future solutions similar to $s$ immediately diminishes. Consequently, our framework captures the natural lifecycle of a task—transitioning from an exploratory phase (no feasible solutions), to an improvable stage (room for optimization), and eventually to a solved state, modeling innovation as the continuous movement of the frontier rather than a static classification.

**Extraction Prompt**

Act as a senior ML engineer and technical writer. You are given two input paths:
1. **research_problem.txt** — a background description of the research problem or competition task (context only).
- Path: `research_problem.txt`
2. **A code project** — the actual implementation that aims to solve this problem (may also include an associated paper, if provided).
- Path: `solution_code/*`

Your task is to read and understand the **entire code repository**, file by file, in order to reconstruct the **solution methodology** comprehensively. Use **research_problem.txt** only as context, but focus primarily on the **code project's solution design** (and paper, if present).

You must produce two outputs:
1. A **summary.md** file (structured textual summary).
2. A **pseudocode.tex** file (LaTeX pseudocode).

Requirements:
- Do **not** include low-level implementation details (file paths, function names, variable names).
- Do **not** add subjective evaluation or speculation.
- Focus on **objective, high-level descriptions** that connect the **problem** with the **solution workflow**.
- Use **research_problem.txt** only as background context, but provide a **detailed and comprehensive explanation of the solution** from the code project (and paper, if available).
- Present content **from overall context → methodology/workflow → abstract details → explicit explanation of how the solution addresses the problem**.
- Ensure **summary.md** and **pseudocode.tex** are consistent and mutually reinforcing.

**Instructions**

1. **Context (from research_problem.txt)** - Summarize the problem background, objectives, and constraints.

2. **Comprehensive Solution Methodology (from the code project and paper, if exists)** - **Workflow**: Provide an end-to-end description (data → model → training → evaluation). - **Modeling Approach**: Describe the architecture at a high level (e.g., modular, hybrid, hierarchical design). - **Data Handling**: Explain how data is ingested, transformed, and used in the pipeline. - **Training Strategy**: Summarize optimization, learning schedules, scaling, and efficiency considerations. - **Evaluation Protocol**: Describe metrics, validation setup, and performance criteria. - **Integration with Research Problem**: Explain clearly and in detail how the solution design directly addresses the stated problem and objectives.

3. **Detailed Abstract Workflow** - Break down the workflow into conceptual steps. - Show the logical flow from data input to problem resolution. - Keep the description neutral and factual.

4. **Outputs** - `summary.md`: Markdown format, structured sections (*Problem Background*, *Objectives*, *Solution Overview*, *Workflow Summary*, *Abstract Details*, *How the Solution Addresses the Problem*). - `pseudocode.tex`: LaTeX pseudocode format, algorithm-style outline of the solution workflow.

**Final Deliverables**
- `./output/summary.md` (comprehensive natural language summary focusing on how the solution addresses the problem)
- `./output/pseudocode.tex` (LaTeX pseudocode capturing the workflow and solution logic)

**Inputs**
- Problem description: `research_problem.txt`
- Code project (and paper, if available): `solution_code/*`

---

**Comparison Prompt**

You are tasked with evaluating the **similarity** between two solutions: the **Agent Solution** and the **Baseline Solution**.

For each of the six evaluation dimensions listed below, do the following:

1. **Score** the similarity on a scale from **0 (Completely Similar)** to **4 (Completely Different)**.

2. **Briefly explain** the rationale behind the score.

**1. Problem Framing & Task Understanding**

- **Task Understanding**: To what extent does the Agent Solution reflect the same understanding of the task as the Baseline Solution?

- **Problem Framing**: How similar is the way the problem is conceptualized or framed between both solutions?

**2. Methodology & Theoretical Basis**

- **Methodology**: How similar are the algorithms, approaches, or techniques used?

- **Theoretical Basis**: How closely aligned are the theoretical foundations (e.g., model assumptions, hypotheses, mathematical frameworks)?

**3. Model Architecture & Implementation**

- **Architecture**: How similar are the core architectures (model types, layers, design patterns)?

- **Implementation**: How similar are the tools, libraries, or programming environments used?

**4. Experiment Design & Validation Methods**

- **Experiment Design**: How aligned are the data setup strategies, such as splitting, preprocessing, or feature handling?

- **Validation Methods**: How similar are the evaluation strategies (metrics, validation splits, cross-validation)?

**5. Algorithm & Optimization**

- **Algorithm Selection**: Are the models or algorithms chosen similar between both solutions?

- **Optimization Techniques**: Are similar optimization methods (e.g., learning rate schedules, regularization, hyperparameter tuning) applied?

**6. Data Processing & Feature Engineering**

- **Data Preprocessing**: Are the methods for data cleaning, scaling, or transformation similar?

- **Feature Engineering**: Are the techniques for feature selection, extraction, or construction similar?

**Instructions for Scoring:**
Use a scale from **0 (Completely Similar)** to **4 (Completely Different)** for each dimension. Provide a short justification for each score.

**Final Output Format (JSON):** { "problem_framing_and_task_understanding": { "score": 1, "justification": "Both solutions define the problem similarly, but the agent interprets the task objective with slight differences." }, "methodology_and_theoretical_basis": { "score": 2, "justification": "The methodologies differ in model complexity, though both are grounded in supervised learning principles." }, "model_architecture_and_implementation": { "score": 1, "justification": "Architectures are similar, but implementation choices (e.g., libraries used) vary." }, "experiment_design_and_validation_methods": { "score": 0, "justification": "Both solutions use identical train/test splits and evaluation

---

metrics." }, "algorithm_and_optimization": { "score": 2, "justification": "Different optimization techniques are used, though the core algorithm is the same." }, "data_processing_and_feature_engineering": { "score": 3, "justification": "The agent applies more advanced feature engineering techniques and a different normalization approach." }, "total_score": 9 }

**Agent Solution:**
{Agent_Solution}

**Baseline Solution:**
{Baseline_Solution}

---

**Encourage Innovation for AIDE Draft Module**

Within reasonable runtime and complexity, you are encouraged to explore more innovative modelling ideas rather than only using the most standard baseline. You may consider non-trivial architectures, custom loss functions, or task-specific feature engineering that could yield better performance, as long as the code remains robust and reproducible.

---

**Encourage Innovation for "AIDE Improve Module"**

When proposing a single actionable improvement, prefer genuinely novel or non-trivial modifications over tiny parameter tweaks, as long as they are still empirically testable. Examples include trying a different model family, changing the training objective, or adding task-specific architectural components, rather than only adjusting one hyper-parameter.

Figure 7: Encourage Innovation Prompt for AIDE

