# OpenReview forum: "InnoGym: Benchmarking the Innovation Potential of AI Agents"
_ICLR.cc/2026/Conference — ICLR 2026 Poster_

### Official Review · Reviewer_UjRg · 2025-10-30

**Soundness:** 2
**Presentation:** 2
**Contribution:** 2
**Rating:** 4
**Confidence:** 2

**Summary:**

The paper introduces InnoGym, a framework and benchmark to quantify the innovation potential of AI agents along two axes: performance gain over the best known solutions and novelty relative to a curated Known Solution Space. The benchmark component iBench comprises 18 Improvable Tasks selected and standardized through a two‑stage curation pipeline. The system component iGym is a unified SDK for long‑horizon, tool‑centric execution. Experiments on 10 tasks evaluate MLAB, CodeAct, and AIDE and analyze time budget, base model, and temperature effects.

**Strengths:**

+ Formalization \(T=(P,S,V,D)\) and metrics \(G,N\) are clear and they enabling evaluation beyond correctnes. This is useful for measuring breakthroughs. Feasibility gating \(C(s)\) ensures novelty is only credited to valid solutions.

+ Dataset curation is well-documented. The authors provide details of two‑stage filtering for resource availability and evaluator quality
with clear visible/hidden split of artifacts.

+ The iGym SDK offers great abstractions for tools/environments, recovery+concurrency, with support for workflow‑ and agent‑mode paradigms.

**Weaknesses:**

- The novelty metric is under‑specified and potentially subjective. \(D(\cdot,\cdot)\) is not clearly defined in Sec. 2.2, and Sec. 4.1 uses to a rubric without a precise equation mapping rubric scores to \(N\). Aggregation/normalization for \(N\) is described qualitatively and thresholds for declaring “innovative success” are not formalized (Sec. 4.1).

- The experimental scope and baseline coverage are limited. Comparisons involve only three agent frameworks (Table 2). Table 2 also includes many missing entries.

- The results lack statistical rigor. Table 2 reports point estimates only (no confidence intervals or multiple seeds). Fig. 5 lacks error bars or variability reporting. No hypothesis tests or bootstrapping are provided for framework comparisons (Sec. 4.2).

- The optimum set notation mixes set and element: “\(S^*\in\arg\max_{s\in S}V(s)\), \(V^*=V(s\in S^*)\)” (Sec. 2.1). The transition from continuous \(D\) (Sec. 2.2) to discrete rubric scoring (Sec. 4.1) lacks a formal link.

- The "Known Solution Space" is said to be comprehensive, but per‑task counts/diversity statistics are not reported (Sec. 4.1). These is also no audit of profile quality (manual verification or inter‑rater agreement) is shown. Since \(S_{\text{known}}\) is finite, coverage can bias novelty (p. 18).

**Questions:**

See weakness.

---

> ### Author Response · Authors · 2025-11-21
> **Response to Reviewer UjRg (Part 1/5, Weakness 1)**
>
> Dear Reviewer UjRg,
>
> We thank you for the very detailed and constructive comments. We are glad that you found the formalization $\mathcal{T}=(P,S,V,D)$, the complementary metrics $G$ and $N$, the two-stage dataset curation, and the iGym SDK abstractions useful. Your suggestions led us to significantly clarify the novelty metric and strengthen the statistical analysis.
>
> A revised draft of our manuscript has been uploaded, with added/modified content marked in green. Below we respond to each concern point-by-point, also indicating changes in the revised manuscript.
>
> ## 1. Response to W1: Novelty metric specification and subjectivity; Link between continuous $D$ and rubric scores
> > **Weakness 1:** The novelty metric is under‑specified and potentially subjective. (D(\cdot,\cdot)) is not clearly defined in Sec. 2.2, and Sec. 4.1 uses to a rubric without a precise equation mapping rubric scores to (N). Aggregation/normalization for (N) is described qualitatively and thresholds for declaring “innovative success” are not formalized (Sec. 4.1).
>
> ### 1.1 Definition: Formal definition of $D$ and $N$
>
> We agree that the earlier version did not spell out the full mapping from rubric scores to a continuous distance. In the revision we now:
>
> * Explicitly define the Agent-as-judge distance $D_{\text{AGENT}}$ in **Appx. E.1, Eq. (4)**: for each dimension $k$ we obtain a discrete score $d_k(s_1,s_2)\in{0,\dots,4}$, normalize by 4, average across dimensions, and rescale to ([0,100]):
> $$D_{\text{AGENT}}(s_1,s_2)=\frac{1}{|K|}\sum_{k\in K}\frac{d_k(s_1,s_2)}{4}\times 100.$$
> * Restate in **Sec. 4.1** how these scores are aggregated over all $h\in S_{\text{known}}$ via the minimum distance and used inside
> $$N(s) = C(s)\cdot\min_{h\in S_{\text{known}}} D_{\text{AGENT}}(s,h).$$
>
> This directly connects the rubric to the continuous novelty metric and removes the ambiguity you pointed out.
>
> ### 1.2 Subjectivity: Validation against human judgments
>
> To address subjectivity, we added experiments in **Appendix E** to test whether our $D$ aligns with human judgments.
>
> * We collected **50 triplets** $(A,B,C)$ from **EquiBench**[1] and created **3 additional triplets** manually. In each triplet, by construction:
>
>   * $A$ and $B$ are **more similar** in method,
>   * $A$ and $C$ are **more different**.
>     So we expect $D(A,C) > D(A,B)$.
> * We then **sampled 11 triplets** and asked human annotators to judge which pair is more different, and compared these judgments against our Agent-as-judge distances.
>
> As reported in **Appendix E (Table 8, Table 9, Table 10, and Table 12)**:
>
> * For the vast majority of triplets, our metric satisfies $D(A,C) > D(A,B)$, matching the known construction.
> * On the randomly sampled subset, the Agent-as-judge distances show good correlation and agreement with human judgments. (Pearson up to ~0.99; triplet-level agreement 75–100%, Table 10 & 12). The table below shows the correlation scores with human judgments, corresponding to Table 10 in the paper.
>   | Dataset           | # Triplets | Pearson r (A,B) | Pearson r (A,C) | Spearman ρ (A,B) | Spearman ρ (A,C) | Triplet agreement |
> |-----|-------:|-------:|----------------:|-----------------:|-----------------:|-------------------|
> | EquiBench (annot.) | 8         | n/a             | 0.84            | n/a              | 0.87             | 6/8 = 75%         |
> | Human triplets   | 3          | 1.00            | 0.99            | 1.00             | 1.00             | 3/3 = 100%        |
>
>
> These results support that, at the granularity relevant to our benchmark, **our instantiation of $D$ is a reasonable and stable proxy for measuring “method-level difference to $S_{\text{known}}$”**. While these studies are necessarily small-scale, they provide concrete evidence that $D_{\text{AGENT}}$ tracks human judgments at both code and method levels rather than reflecting arbitrary model preferences.
>
>
> ### 1.3 Thresholds for “innovative success”.
> We also agree that the connection between $N$ and phrases such as “innovative success” should be clarified.
>
> We highlight that the novelty score is **inherently relative rather than absolute to $S_{\text{known}}$** for each task. Because different tasks have different prior solution sets and diversity (quantified in Table 3), a single global numeric threshold such as “$N>60$” would not be comparable across tasks.
>
> Conceptually, the framework views “innovative success” as requiring *both* a performance improvement and some methodological departure from prior work. In practice we reason about *regimes* of innovation—for example:
>
> * high $G$, low $N$: strong performance gains via conventional methods;
> * high $G$, high $N$: gains achieved by methodologically distinct ideas;
> * moderate $G$, high $N$: exploratory but promising directions.
>
> Finally, the only strict gating is **feasibility**: novelty is awarded only when $C(s)=1$. Infeasible or invalid solutions always receive $N(s)=0$, regardless of how different they look in natural language.

---

> ### Author Response · Authors · 2025-11-21
> **Response to Reviewer UjRg (Part 2/5, Weakness 2)**
>
> ## 2. Response to W2: Experimental scope, baseline coverage, and missing entries in Table 2
> > **Weakness 2:** The experimental scope and baseline coverage are limited. Comparisons involve only three agent frameworks (Table 2). Table 2 also includes many missing entries.
>
> ### 2.1 Choice of three agent frameworks.
> Our goal in this first version of InnoGym was to cover **diverse design paradigms** rather than many closely related variants. Following MLE-Bench[2], we selected three agent scaffolds that make qualitatively different design choices for research agents (described in **Appx. D.1**):
>
> * **MLAB**: a ReAct-style, workflow-oriented agent that plans in language and manipulates an ML workspace (read/edit files, run training scripts, inspect logs).
> * **CodeAct**: a code-centric framework in which every action is a short executable Python program that directly calls libraries, runs shell commands, and debugs itself through iterative execution.
> * **AIDE**: a tree-search–based agent that starts from an initial solution and repeatedly proposes code edits, evaluates them, and branches on promising variants.
>
> Together, these three scaffolds cover workflow-style planning, code-centric action spaces, and search-based exploration, giving a diverse slice of current agent designs for innovation-oriented tasks.
>
> ### 2.2 Meaning of missing entries and their role in analysis
> We agree that the “/” entries in Table 2 should be clearly explained. The revised text in Sec. 4.1 and Appx. D.2 now states that:
>
> * A “/” means that **all three runs for that task–agent configuration failed** to produce a valid submission within the 12-hour budget, i.e., success rate 0% and $C(s)=0$.
> * For statistical analysis, we adopt a **pessimistic imputation** scheme in Appx. D.2: for every such failure case, we assign the minimum normalized ratio $R=-1$ and novelty $N=0$. This ensures that failure cases are explicitly included when computing macro-averages and in hypothesis tests, rather than being silently ignored.
>
> We further report per-task submission success rates over three runs in **Table 6 (Appx. D.3)**. Notably, on two tasks (CDML and PTTALC) *no* framework managed a valid submission, and several other tasks have low success rates.  Combined with the performance gaps in Table 2, this supports our claim that iBench focuses on difficult, long-horizon, high-value problems that current agents struggle with, rather than on “easy wins”.

---

> ### Author Response · Authors · 2025-11-21
> **Response to Reviewer UjRg (Part 3/5, Weakness 3)**
>
> ## 3. Response to W3: Statistical rigor: confidence intervals, seeds, and variability in Fig. 5
> > The results lack statistical rigor. Table 2 reports point estimates only (no confidence intervals or multiple seeds). Fig. 5 lacks error bars or variability reporting. No hypothesis tests or bootstrapping are provided for framework comparisons (Sec. 4.2).
>
> We appreciate the request for more rigorous statistics and have substantially strengthened this part of the paper.
>
> **(a) Multiple runs and error bars in Fig. 5.**
> Each configuration in the analysis experiments is now run **three times**. In the updated version of **Fig. 5(a)** (time budget) and **Fig. 5(b)** (base model), we plot the mean over three runs with error bars, and we updated the caption to explicitly state that each setting is “run 3 times”. This quantifies the variability in these ablations instead of showing single-point estimates.
>
> **(b) Macro-averages and bootstrap confidence intervals for Table 2.**
> To address the lack of confidence intervals in Table 2, we added **Appx. D.2**, which introduces a task-level bootstrap analysis:
>
> * Under the pessimistic imputation described above, every framework obtains a ratio and novelty score on each of the 10 main tasks.
> * For each agent framework $f$ and metric $M\in{R,N}$, we compute the macro-average over tasks and estimate **95% non-parametric bootstrap confidence intervals** with $B=10{,}000$ resamples. The results are reported in **Table 4**, showing the mean and confidence interval for each framework.
>
> This provides uncertainty estimates instead of bare point estimates.
>
> **(c) Paired comparisons across frameworks.**
> We also perform paired comparisons across frameworks in **Table 5**: for each pair of agent frameworks $(f,f')$ and metric $M$, we bootstrap the task-level difference $\Delta M_{f,f'}$ and report 95% confidence intervals and approximate two-sided bootstrap $p$-values. The results show that:
>
> * MLAB has a significantly higher macro-average ratio $R$ than both CODEACT and AIDE (statistically significant according to the bootstrap test).
> * For novelty $N$, MLAB also has the highest mean, but differences between frameworks are not statistically significant at the 0.05 level given the small number of tasks. We therefore interpret the $N$ results as *descriptive trends* and use $R$ as the main metric for formal cross-framework comparisons.
>
> **(d) Seeds and computational budget.**
> Due to the high computational cost of long-horizon agents (up to 12 hours per run), we cannot afford a large number of seeds for every configuration. Instead, we:
>
> * run each configuration three times,
> * report error bars for ablations, and
> * rely on **bootstrap over tasks** to quantify uncertainty and compare frameworks.
>
> We explicitly note this computational limitation in the text and view our combination of multi-run error bars, bootstrap confidence intervals, and paired tests as a reasonable compromise between rigor and feasibility at this scale.

---

> ### Author Response · Authors · 2025-11-21
> **Response to Reviewer UjRg (Part 4/5, Weakness 4)**
>
> ## 4. Response to W4: Optimum set notation and link between continuous $D$ and discrete rubric scores
> > **Weakness 4:** The optimum set notation mixes set and element: “(S^\in\arg\max_{s\in S}V(s)), (V^=V(s\in S^*))” (Sec. 2.1). The transition from continuous (D) (Sec. 2.2) to discrete rubric scoring (Sec. 4.1) lacks a formal link.
>
> We appreciate the pointer to the notation issue in Sec. 2.1. In the revised version, the optimal value and optimal set are defined using standard set notation as:
> $$V^* = \max_{s\in S,,C(s)=1} V(s), \quad S^* = \{s\in S \mid C(s)=1,; V(s)=V^*\},$$
> which avoids mixing elements and sets.
>
> Regarding the transition from the abstract, continuous distance $D$ in Sec. 2.2 to the rubric-based scoring in Sec. 4.1:
>
> * Sec. 2.2 now explicitly notes that $D$ is instantiated as an Agent-as-judge score in our experiments, with details deferred to Appx. E.
> * Appx. E.1 provides the formal equation mapping discrete rubric scores to a continuous distance $D_{\text{AGENT}}$, which is then plugged into the general definition of $N$ introduced in Sec. 2.2.
>
> This makes the link between the framework-level definition and the concrete implementation precise. As a side benefit, the rubric formulation keeps the framework flexible: different tasks can adopt different rubrics while remaining compatible with the same $(G,N)$ formulation.

---

> ### Author Response · Authors · 2025-11-21
> **Response to Reviewer UjRg (Part 5/5, Weakness 5)**
>
> ## 5. Response to W5: Coverage and quality of the known solution space $S_{\text{known}}$
> > The "Known Solution Space" is said to be comprehensive, but per‑task counts/diversity statistics are not reported (Sec. 4.1). These is also no audit of profile quality (manual verification or inter‑rater agreement) is shown. Since $S_{\text{known}}$ is finite, coverage can bias novelty (p. 18).
>
> We agree that calling $S_{\text{known}}$ “comprehensive” was too strong. In the revision, we:
>
> **(a) Remove overly strong wording and add explicit statistics.**
> We no longer describe $S_{\text{known}}$ as comprehensive. Instead, we emphasize the **construction process and measured diversity**:
>
> * **Appx. F.1** details how we build $S_{\text{known}}$: we collect candidate solutions from leaderboards, papers, and public code repositories, augment them using search tools, and have multiple team members independently search and curate candidates before merging. Each candidate must pass a **reproducibility filter** inside iGym (executability + score consistency) to be retained.
> * **Table 3** now reports, for each task, the number of retained reference solutions $|S_{\text{known}}|$ and a diversity statistic $Div(T)$, defined as the average pairwise $D_{\text{AGENT}}$ distance among reference solutions (Rao’s quadratic diversity with uniform weights). See Appendix C for more details.
>
> This gives a concrete sense of both the *size* and *methodological spread* of $S_{\text{known}}$ per task.
>
> **(b) Profile quality and potential bias.**
> First, we evaluate the alignment between the novelty evaluator $N$ and human assessments in Appendix E, demonstrating strong agreement, and provide the diversity statistics for $S_{\text{known}}$ in Appendix C. We do audit quality by:
>
> * verifying that each profile reproduces the reported performance under our standardized evaluator;
> * standardizing each solution into a structured description using a fixed extraction prompt before feeding it to the novelty evaluator.
>
> Then we emphasize that novelty is always measured **relative to a finite** $S_{\text{known}}$, which can bias scores when prior coverage is limited. To mitigate this and make the benchmark extensible:
>
> * All reference profiles, prompts, and evaluation scripts will be released so that the community can *extend* $S_{\text{known}}$ or plug in alternative novelty metrics.
> * Any extension of $S_{\text{known}}$ can only **decrease** novelty scores (by adding more nearby references) and does so *equally* for all agents on that task, preserving fairness of relative comparisons.
> * For tasks with inherently few prior solutions (e.g., highly specialized competitions), limited $S_{\text{known}}$ reflects the true scarcity of existing approaches; in such settings, both human and agent judgments of novelty are unavoidably based on partial prior knowledge.
>
> Overall, our goal is not to estimate an absolute, universe-wide notion of novelty, but to provide a **transparent, reproducible, and task-relative** measure of methodological distance to known solutions. We believe the additional statistics and documentation make this dependence explicit and help readers interpret $N$ appropriately.
>
> ---
>
> Once again, we thank you for the thoughtful and constructive feedback. We believe the clarifications, additional experiments, and textual revisions described above substantially strengthen the paper and make our design choices and limitations more transparent.
>
> **Please let us know if you have any further questions. If you find that our response addresses some of your concerns, would you kindly consider raising your rating score for our paper? We greatly appreciate your consideration.**
>
> ---
>
> ## Reference
> [1] EquiBench. ACL2025. https://arxiv.org/abs/2502.12466.
>
> [2] MLEBench. ICLR2025. https://arxiv.org/abs/2410.07095
>
> ---
>
> Best Regards,
>
> Authors of InnoGym

---

> ### Author Response · Authors · 2025-11-27
> **Kindly Reminder**
>
> Dear reviewer UjRg,
>
> We highly appreciate the constructive comments and insightful suggestions you have offered for our work. As the discussion period has only a few days left, in order for us to have sufficient time to address any additional questions you may have, we kindly encourage you to engage in the ongoing discussion and share any further insights or clarifications you may have.
>
> **If you find that our response addresses some of your concerns, would you kindly consider raising your rating score for our paper?**
>
> Thank you very much for your time and consideration. We look forward to hearing from you soon.
>
> Best Regards,
>
> Authors of InnoGym

---

### Official Review · Reviewer_Rk9e · 2025-10-31

**Soundness:** 3
**Presentation:** 3
**Contribution:** 2
**Rating:** 4
**Confidence:** 3

**Summary:**

This paper proposes InnoGym, a benchmark that aims to evaluate the innovation potential of AI agents beyond just producing the correct answers. Agents are evaluated based on both performance gains and novelty (methodological differences). The authors include 18 engineering tasks with iGym as the execution environment. Experiments show the different behavior across different methods. Further analysis reveals the impact of prior knowledge, execution time, base models, and sampling temperature.

**Strengths:**

1.	Evaluating the model performance on novelty, besides performance gain, is interesting and important. In the context of the benchmark, since the solution space can be grounded with known solutions, the evaluation of novelty can be quantified.

2.	This paper is well-written and easy to follow, with clear figurative illustrations.

3.	The authors show good comparison with existing benchmarks in Table 1 to consolidate the motivation.

4.	The analysis, e.g., prior solution, is well-executed.

**Weaknesses:**

1.	The definitions in Section 2.3 are very strong and not well-grounded. For example, SWE-Bench is defined as a “solved problem”. However, the optimal solutions to the bugs are clearly not defined or estimated. It is also an unsolved/improvable task for agents. On the other hand, for improvable tasks, how is achieving a new state-of-the-art performance defined as novelty? For example, this might be achieved by simply extending the training or using more in-domain data.

2.	The evaluation of novelty is much dependent on the distance function. However, there is not enough detail about how the function is implemented. Is it done by the Comparison prompt shown in B.2? It is unclear what distance function can measure the similarity in solutions to extract a novelty score. For example, two code files might be semantically dissimilar but have the same logic. See BRIGHT (https://arxiv.org/abs/2407.12883) and EquiBench (https://arxiv.org/pdf/2502.12466) for further understanding of this question.


3.	Extending from 2, there is no human grounding for the novelty/differences in the solution. It remains unclear about the qualification of the LLM-as-a-judge. The authors can either improve the robustness by breaking the solution down into tools/methods and showing what patterns are captured by the judge. Alternatively, the authors can show the correlation between judges and humans.


4.	Do we actually need novelty in some ML competition tasks at this stage, especially when all methods have very low scores in Table 2? The quality of the novelty evaluation is bounded by the candidate solutions. However, the human candidate solution set is not acquired with a novelty objective. Simple methods that are executed well but not necessarily novel can be a good solution for some of these tasks

5.	Given that the current definition of novelty is more like diversity, which covers part of the whole space of innovation (e.g., not covering creating new research direction), it would be good if the authors could show some pragmatic usefulness of the current definition, e.g., diverse solutions can make a better ensemble method.

**Questions:**

1. One small concern is that there is no specific instruction on novelty given to the model / agentic framework. I am wondering how such instruction would influence the novelty scores.

2. The citations to MLAB, CodeAct, and AIDE seem missing.

---

> ### Author Response · Authors · 2025-11-21
> **Response to Reviewer Rk9e (Part 1/6, Weakness 1)**
>
> Dear Reviewer Rk9e,
>
> We thank you for the thoughtful and constructive feedback. We are encouraged by your positive assessment of our work, particularly your recognition of **the importance of evaluating novelty alongside performance**, **the rigor of our benchmark comparisons and analysis**, and **the overall clarity of our presentation.**
>
> A revised draft of our manuscript has been uploaded, with added/modified content marked in green. Below, we address your concerns point by point.
>
> ---
>
> ## 1. Response to Weakness 1
>
> > Weakness 1:
> > `W1.1`: The definitions in Section 2.3 are very strong and not well-grounded. For example, SWE-Bench is defined as a “solved problem”. However, the optimal solutions to the bugs are clearly not defined or estimated. It is also an unsolved/improvable task for agents.
> > `W1.2`: On the other hand, for improvable tasks, how is achieving a new state-of-the-art performance defined as novelty? For example, this might be achieved by simply extending the training or using more in-domain data.
>
> ### 1.1 Response to W1.1: On the task taxonomy in Section 2.3 and the SWE-Bench example
>
> We sincerely thank the reviewer for this insightful comment. You have pinpointed a crucial aspect of our taxonomy that we should have made more explicit: that **these classifications are intentionally relative and dependent on the specific, human-defined goal for a task.** This is, in fact, a central tenet of our framework, and we appreciate the opportunity to clarify it. And we have added a detailed definition of the task in Appendix C.1.
>
> To ground our discussion, let's revisit the core definitions from Section 2.3. Our taxonomy is defined **per concrete task instance and relative to known human solutions**, not for an entire benchmark with respect to agents:
> - For a given instance $P$, if a known solution $s \in S_{\text{known}}$ passes the validator $C$ and achieves the optimal objective $V^*$, it is a **solved problem.**
> - If s \in S_{\text{known}} passes $C$ but does not reach $V^*$, it is an **improvable task.**
> - If no $s \in S_{\text{known}}$ passes $C$, it is an **exploratory problem.**
>
> **Applying This Definition to SWE-Bench.**
>
> Your point about the "optimal solution" for SWE-Bench is excellent because it highlights exactly **why our framework adopts a precise, operational definition.** Instead of seeking a universal, abstract notion of optimality, our framework defines it **strictly in relation to the task's explicit validator and objective.**
> By this definition, a standard SWE-Bench instance is a **"solved problem"**:
> - The objective $V^*$ is to pass the provided test suite.
> - The existing human patch $s_{\text{known}}$ is a known solution that accomplishes this.
>
> Because the defined goal has been fully met, the instance fits our definition of "solved." This is not a claim that a better implementation is impossible, but a classification based on the stated goal. As you correctly reasoned, if we were to change the objective to "fix the bug with minimal edits," the task would immediately become "improvable," demonstrating the deliberate flexibility of our taxonomy.
>
>
> ### 1.2 Response to W1.2: Can a simple yet highly effective method be considered innovative within the Innogym?
>
> In our framework, innovation has **two independent axes**:
>
> * **Performance gain** $G(s) = V(s) - V^*_{\text{known}}$, and
> * **Novelty** $N(s)$, defined as the minimum method-level distance to $S_{\text{known}}$:
> $N(s) = C(s)\cdot \min_{h\in S_{\text{known}}} D_{\text{AGENT}}(s,h),$
>   where $D_{\text{AGENT}}$ is described in detail below and in Appx. E.1.
>
> Achieving a new SOTA performance **does not automatically imply high novelty**: if a solution closely follows an existing high-performing pipeline (e.g., same modeling paradigm and training recipe, but more epochs or slightly more in-domain data), the comparison rubric assigns low per-dimension scores, resulting in a **high $G$** but **low $N$**. Conversely, a solution that changes major methodological dimensions (e.g., model family, optimization strategy, or problem framing) will score higher in $N$, regardless of whether it reaches SOTA.
>
> From an empirical standpoint, we classify the following three categories as innovative solutions:
> - High-$G$, Low-$N$ (simple yet effective);
> - High-$G$, High-$N$ (novel and effective);
> - Low-$G$, High-$N$ (where $G$ approaches zero, representing novel and comparative approaches).

---

> ### Author Response · Authors · 2025-11-21
> **Response to Reviewer Rk9e (Part 2/6, Weakness 2, Weakness 3)**
>
> ## 2. Response to W2 and W3: On the distance function $D$, semantic equivalence, and agent-as-judge
>
> > **Weakness 2:** The evaluation of novelty is much dependent on the distance function. However, there is not enough detail about how the function is implemented. Is it done by the Comparison prompt shown in B.2? It is unclear what distance function can measure the similarity in solutions to extract a novelty score. For example, two code files might be semantically dissimilar but have the same logic. See BRIGHT and EquiBench for further understanding of this question.
>
> > **Weakness 3:** Extending from 2, there is no human grounding for the novelty/differences in the solution. It remains unclear about the qualification of the LLM-as-a-judge. The authors can either improve the robustness by breaking the solution down into tools/methods and showing what patterns are captured by the judge. Alternatively, the authors can show the correlation between judges and humans.
>
> We fully agree that the distance function $D$ is central. We apologize for not making its instantiation prominent enough in the main text. We now already provide a detailed description and human validation in Appx. E; we also have moved a condensed version into the main paper (Sec. 2.2 / 4.1) in the revised version. Below is a brief summary.
>
> ### 2.1 Concrete implementation of $D$: method-level, agent-as-judge, and two-stage pipeline
>
> In all experiments, we instantiate $D$ as a **method-level agent-as-judge distance** $D_{\text{AGENT}}$, designed to be robust to superficial code changes and sensitive to methodological differences (similar in spirit to EquiBench/BRIGHT):
>
> 1. **Extraction (Appx. E.1 & G.1).**
>    For each solution (historical or agent-generated), we run an extraction prompt over the full repository. The model produces two normalized artifacts:
>
>    * `summary.md`: a structured description of the problem, data pipeline, model architecture, and training procedure.
>    * `pseudocode.tex`: LaTeX-style pseudocode capturing the core algorithmic steps.
>
>    This step intentionally strips away variable names, file layout, and other stylistic noise.
>
> 2. **Comparison (Appx. E.1 & G.2).**
>    Given two solutions $s_1, s_2$ (with their summary + pseudocode), a second prompt asks the model to act as a reviewer and score their **method-level similarity** along a fixed set of dimensions $K$ (e.g., problem framing, methodology/theoretical basis, model architecture and implementation, data processing and feature engineering, optimization, evaluation protocol). For each dimension $k\in K$, the model outputs:
>    $$d_k(s_1,s_2) \in {0,1,2,3,4}$$
>    where 0 ≈ “essentially the same” and 4 ≈ “completely different paradigm” for that aspect.
>
>    We then aggregate:
>    $$D_{\text{AGENT}}(s_1,s_2) = \frac{1}{|K|}\sum_{k\in K}\frac{d_k(s_1,s_2)}{4}\times 100,$$
>    yielding a **0–100** method-distance. $N(s)$ is the minimum of this distance over $S_{\text{known}}$.
>
> Thus, $D_{\text{AGENT}}$ is not a raw token-level, but a rubric-driven measure of **solution methodology**, which is exactly the level at which BRIGHT and EquiBench motivate semantic equivalence. We will explicitly mention this connection in the revised paper.
>
> ### 2.2 Human validation on EquiBench code-level equivalents
>
> Motivated by your comment and EquiBench, we first test whether $D_{\text{AGENT}}$ ignores purely cosmetic edits while reacting to algorithmic changes (Appx. E.2–E.3).
>
> * On **50 EquiBench triplets** $(A,B,C)$, where by construction $B$ is a cosmetic variant of $A$ (same algorithm) and $C$ is an algorithmically different variant:
>
>   * Mean distances (Table 8):
>
>     * $D_{\text{AGENT}}(A,B) = 1.00$
>     * $D_{\text{AGENT}}(A,C) = 9.75$
>       (0–100 scale).
> * On a **subsample of 8 triplets**, three CS graduate students rate the distances $Human(A,B)$ and $Human(A,C)$ on the same 0–100 scale (Appx. G.2 rubric). As shown in Table 9:
>
>   * $Human(A,C)$ averages 16.66, $Human(A,B)$ ≈ 0.
>   * $D_{\text{AGENT}}(A,C)$ averages 15.63, $D_{\text{AGENT}}(A,B)$ ≈ 1.04.
>
> The table below shows the correlation scores with human judgments, corresponding to Table 10 in the paper.
>   | Dataset           | # Triplets | Pearson r (A,B) | Pearson r (A,C) | Spearman ρ (A,B) | Spearman ρ (A,C) | Triplet agreement |
> |--|---:|--:|--:|--:|--:|--|
> | EquiBench (annot.) | 8  | n/a             | 0.84            | n/a              | 0.87             | 6/8 = 75%         |
>
> From Table 10, we can observe that:
>
> * For the 8 human-annotated EquiBench triplets, correlations ($A,C$ pairs only, since $human(A,B)$ are all 0):
>
>   * Pearson $r = 0.84$, Spearman $\rho = 0.87$.
> * Triplet agreement (whether C is judged more novel than B relative to A): **6/8 = 75%**.
>
> These results show that $D_{\text{AGENT}}$ (1) treats cosmetic variants as essentially zero-distance and (2) tracks human judgments closely on which variant is more novel relative to a base solution.

---

> ### Author Response · Authors · 2025-11-21
> **Response to Reviewer Rk9e (Part 3/6, Weakness 2, Weaknees 3)**
>
> ### 2.3 Human validation on cross-domain method triplets
>
> We also construct triplets from real research methods (e.g., baseline A, variant B in the same paradigm, and clearly paradigm-shifting method C in a sub-area such as long-context modeling). Domain-expert annotators and our agent-as-judge both consistently rank C as more novel relative to A, with similar scores on a 0–100 scale. See Appendix E.4 for details. The following result shows the $D$ being consistent with human expert judgement.
>
> | Dataset  | # Triplets | Pearson r (A,B) | Pearson r (A,C) | Spearman ρ (A,B) | Spearman ρ (A,C) | Triplet agreement |
> |--|---:|--:|--:|--:|--:|--|
> | Human triplets   | 3 | 1.00            | 0.99            | 1.00             | 1.00             | 3/3 = 100%        |
>
> We do not claim LLM judgments are equivalent to human experts, but **method decomposition + human agreement on multiple triplets** provides evidence that agent-as-judge is a reasonable, grounded proxy at our level of abstraction. We agree that there is room to make $D$ more “intelligent”. Here, we outline a direction for **future work**:
>
> * Design a **specialized agent** whose job is to:
>
>   * search the literature and code repositories,
>   * construct and update $S_{\text{known}}$,
>   * and compare new solutions against this enlarged knowledge base.
> * This would give a more powerful instantiation of $D$, still within our general framework.
>
> We stress that this is beyond the scope of the current paper and is left as **future work**, but it fits naturally into our design: it would simply be a more advanced, agent-powered version of $D$.

---

> ### Author Response · Authors · 2025-11-21
> **Response to Reviewer Rk9e (Part 4/6, Weakness 4)**
>
> ## 3. Response to Weakness 4: On the necessity and role of novelty, especially when scores are low
>
> > **Weakness 4:** Do we actually need novelty in some ML competition tasks at this stage, especially when all methods have very low scores in Table 2? The quality of the novelty evaluation is bounded by the candidate solutions. However, the human candidate solution set is not acquired with a novelty objective. Simple methods that are executed well but not necessarily novel can be a good solution for some of these tasks.
>
> We fully agree that **simple, well-executed methods are valuable**. In InnoGym, the primary leaderboard is still based on **performance gain $G$**: high-$G$, low-$N$ solutions are ranked at the top and are desirable in practice.
>
> Novelty plays a different role:
>
> 1. **Characterizing the known solution space.**
>    For each task $T$, we construct $S_{\text{known}}(T)$ from top human/leaderboard solutions and literature (Appx. F), and quantify their method-level spread via:
>    $$\text{Div}(T) = \frac{2}{m(m-1)}\sum_{1\le i<j\le m} D_{\text{AGENT}}(h_i,h_j),$$
>    where $h_i\in S_{\text{known}}(T)$. This indicates how much of the method space is already covered by strong human solutions. See Appendix C for more details.
>
> 2. **Diagnosing agent behavior, not penalizing good engineering.**
>    Given this backdrop, measuring $N(s)$ helps us distinguish between:
>
>    * agents that **re-implement** known paradigms $high (G), low (N)$, and
>    * agents that **explore qualitatively new approaches** (potentially high $N$, with varying $G$).
>
>    We do **not** penalize high-$G$, low-$N$ solutions in the main metric; novelty is used primarily as an analysis axis (e.g., Fig. 4–5), to understand whether agents are stuck in imitation or genuinely exploring.
>
> 3. **Understanding exploration–exploitation dynamics.**
>    Even when absolute scores are low, trajectories in the $(G,N)$ plane are informative. For example, in Fig. 4, we see agents initially move to high novelty (exploration) and later converge toward lower novelty with higher performance (exploitation). Without $N$, this dynamic would be invisible.
>
> We will clarify in the paper that novelty is intended as a **diagnostic and analysis tool**, complementary to performance, rather than a replacement for simple, strong baselines.

---

> ### Author Response · Authors · 2025-11-21
> **Response to Reviewer Rk9e (Part 5/6, Weakness 5)**
>
> ## 4. Response to W5: On “novelty ≈ diversity” and practical usefulness (e.g., ensembles)
>
> > **Weakness 5:** Given that the current definition of novelty is more like diversity, which covers part of the whole space of innovation (e.g., not covering creating new research direction), it would be good if the authors could show some pragmatic usefulness of the current definition, e.g., diverse solutions can make a better ensemble method.
>
> We agree our current $N$ is best described as **methodological diversity relative to the known solution set**. It mainly captures:
>
> - whether a solution uses a method that is similar to or different from existing human methods **within the same task definition**.
>
> It does **not** capture broader aspects of innovation such as creating entirely new research directions or redefining the task itself. We make this clearer in the revised paper and explicitly list it as a limitation.
>
> We also agree that diverse, high-performing solutions are natural candidates for better ensembles/portfolios. Due to ensemble feasibility and engineering cost, we did not yet run full ensemble experiments, but we now highlight this as **an important piece of future work**:
>
> - The performance of current solutions is relatively poor, making it difficult to perform ensembling effectively. In classification problems with many categories, for example, it is hard to achieve strong consistency across multiple solutions because their individual performance is weak. Additionally, for non-classification tasks, such as generating a specific ordering or permutation, ensembling the results is particularly challenging
>
> - On appropriate tasks, one can directly compare ensembles built from **high $G$ + high $N$** solutions versus those from **high $G$ + low $N$** solutions, to test whether our novelty metric adds downstream value.
>
> Even without ensembles, we already find $N$ useful in practice: it helps diagnose agent behavior, visualize search trajectories, and prioritize diverse candidates for human inspection, as shown in response to Weakness 4. We see InnoGym as providing the infrastructure to make such diversity-aware studies straightforward.

---

> ### Author Response · Authors · 2025-11-21
> **Response to Reviewer Rk9e (Part 6/6, Question 1, Question 2)**
>
> ## 5. Response to Q1: On not explicitly instructing agents to be “novel”
>
> > Q1: One small concern is that there is no specific instruction on novelty given to the model / agentic framework. I am wondering how such instruction would influence the novelty scores.
>
> In all main results, agents are only instructed to **maximize the competition objective**, without any mention of novelty. This is intentional:
>
> - It keeps both $G$ (performance gain) and $N$ (novelty) as **external evaluation metrics**, allowing us to analyze current agents without entangling the benchmark with a particular “be creative” training objective.
> - It lets us first ask: *How much method diversity do existing agents naturally produce when optimizing only for performance?*
>
> To directly examine the effect of encouraging novelty, we add a new experiment in **Appx. D.3, “Encourage Novelty Explicitly”**, where we modify the prompts of the AIDE agent on three tasks: BETTL(sleep), OAG, and CirclePacking. We compare:
>
> - **Baseline:** standard competition-oriented prompting.
> - **Innovative:** prompts that explicitly encourage “creative / innovative” methods while still asking for high scores (prompt shown in Fig. 7).
>
> The results (shown in **Table 7**) are:
>
> | Task           | Gain (Ratio) Baseline | Gain (Ratio) Innovative | Novelty Baseline | Novelty Innovative |
> |----------------|------------------------|-------------------------|------------------|--------------------|
> | BETTL(sleep)   | -53.62 (-0.77)         | -54.56 (-0.79)          | 54.17            | 58.33             |
> | OAG            | -29.87 (-0.36)         | -31.76 (-0.38)          | 70.83            | 50.00             |
> | CirclePacking  | -0.25 (-0.09)          | -0.37 (-0.14)           | 35.33            | 58.33             |
>
> We find:
>
> - On **all three tasks**, explicitly encouraging “innovative” solutions **reduces performance gain** (more negative Gain and Ratio values).
> - The effect on **novelty** is task-dependent but can be large:
>   - For **CirclePacking**, novelty increases markedly from 35.33 to 58.33.
>   - For **BETTL(sleep)**, novelty increases modestly (54.17 → 58.33).
>   - For **OAG**, novelty actually decreases (70.83 → 50.00), suggesting that simply asking for “innovation” does not guarantee more diverse methods on every task.
>
> We therefore observe a clear **exploration–exploitation trade-off**:
>
> - Stronger “be innovative” signals (through prompts) can push agents toward more novel regions of the method space (especially on some tasks),
> - but under current models and frameworks, this **comes at a consistent cost in solution quality**.
>
> Combined with the temperature ablation in the main text (Fig. 5(c)—where higher temperatures also increase novelty at the expense of performance), these results indicate that our novelty metric $N$ is sensitive to how exploratory the agent behaves, and that simply asking for more novelty is not enough: future agents will need to **coordinate creativity with robustness** rather than maximizing one at the expense of the other.
>
> ---
>
> ## 6. Response to Q2: On missing citations for MLAB, CodeAct, and AIDE
>
> > Q2: The citations to MLAB, CodeAct, and AIDE seem missing.
>
> Thank you for catching this. In the revised version, we have added the missing citations for **MLAB**, **CodeAct**, and **AIDE** where they are first introduced in Section 4.1, and we briefly summarize their designs in Appendix D.1.
>
> ---
>
> Once again, we thank you for the insightful feedback. **Please let us know if you have any further questions. If you find that our response addresses some of your concerns, would you kindly consider raising your rating score for our paper? We greatly appreciate your consideration.**
>
> ---
>
> ## Reference
>
> [1] EquiBench. ACL2025. https://arxiv.org/abs/2502.12466.
>
> ---
>
> Best Regards,
> Authors of InnoGym

---

> ### Author Response · Authors · 2025-11-27
> **Kindly Reminder**
>
> Dear reviewer Rk9e,
>
> We highly appreciate the constructive comments and insightful suggestions you have offered for our work. As the discussion period has only a few days left, in order for us to have sufficient time to address any additional questions you may have, we kindly encourage you to engage in the ongoing discussion and share any further insights or clarifications you may have.
>
> **If you find that our response addresses some of your concerns, would you kindly consider raising your rating score for our paper?**
>
> Thank you very much for your time and consideration. We look forward to hearing from you soon.
>
> Best Regards,
>
> Authors of InnoGym

---

### Official Review · Reviewer_ELHx · 2025-10-31

**Soundness:** 3
**Presentation:** 2
**Contribution:** 3
**Rating:** 6
**Confidence:** 4

**Summary:**

This paper proposes InnoGym, a benchmark and execution framework to rigorously evaluate AI beyond correctness, to include ‘innovation’. The authors formalize innovation as a combination of performance gain over the best-known solutions and methodological novelty, and introduce an evaluation pipeline grounded in this definition. They curate 18 improvable real-world scientific and engineering tasks, standardize executors and validators. Relatedly, the authors also propose iGym, a unified tool-and-execution environment to support agent evaluation on its innovation. Empirical studies across LLM agents show that while some systems produce novel solutions, they rarely surpass human baselines and often fail due to robustness limitations. An insight here is that novel ideas alone are insufficient, i.e. the key bottleneck for current agentic systems is reliably executing and scaling those ideas into verified performance improvements.

**Strengths:**

The creativity evaluation framework, suite of tasks and gym could be a helpful resource for the AI community.

The approach of evaluating LLM and agents’ creativity through their generation solution and performance is sound and intuitive.

**Weaknesses:**

The creativity evaluation framework seems to involve heavy amount of computation for each candidate solution as an individual AI model as training and evaluation have to be done for each of them.

Several key details are missing in the main text of the paper which can weaken its clarity and reproducibility, for example, what exactly are the distance functions used for each task (seems to be B.2 COMPARISON PROMPT but not mentioned in main text), how are the tasks processed to be use for creativity measure (e.g. some of the tasks are classification based) and meaning of the three methods used in the experiments (e.g. MLAB, CodeAct, AIDE).

**Questions:**

Formatting in page 14 seems off.

Please see weaknesses.

---

> ### Author Response · Authors · 2025-11-21
> **Response to Reviewer ELHx (Part 1/3, Weakness 1)**
>
> Dear Reviewer ELHx:
>
> We thank you for recognizing the value of our creativity evaluation framework, the comprehensive task suite, and the iGym platform. We appreciate the positive feedback on our sound and intuitive approach to evaluating innovation beyond correctness.
>
> A revised draft of our manuscript has been uploaded, with added/modified content marked in green. Below, we respond to each concern in turn.
>
> ## 1. Response to W1: Computational cost of evaluating candidate solutions
> > **W1:** The creativity evaluation framework seems to involve heavy amount of computation for each candidate solution as an individual AI model as training and evaluation have to be done for each of them.
>
> We address this from two perspectives:
>
> **(a) Novelty evaluation is lightweight and cacheable.**
> The *novelty* evaluation introduces only a small, controllable overhead:
>
> * For each candidate solution $s$, iGym runs a single **extraction step**: a fixed prompt converts the solution into a compact profile (pseudocode + short natural-language description). This profile is **cached**. See Appendix E.1 for details.
> * To compute novelty, we compare this cached profile against each reference in $S_{\text{known}}$ using the comparison prompt (Appendix G.2). This yields rubric scores that are aggregated into a continuous distance $D_{\text{AGENT}}$, and then into $N(s) = C(s)\cdot \min_{h\in S_{\text{known}}} D_{\text{AGENT}}(s,h)$.
>
> Thus, for a given solution we incur **one extraction + $|S_{\text{known}}|$ comparisons**, all of which are short LLM calls. If $S_{\text{known}}$ is later extended, we reuse the cached profile and only run additional comparisons to new references; we never re-run extraction or any model training.
>
> Given that a high-quality agent solution may take hours of computing to produce, we believe it is reasonable and even necessary to allocate some time for a careful evaluation of its creativity, just as human reviewers spend substantial effort to understand and judge a new idea. Our design tries to keep this overhead modest and scalable (linear in $|S_{\text{known}}|)$, while still giving a meaningful, method-level notion of novelty.
>
> **(b) No extra requirement to train models during evaluation.**
> The creativity evaluator itself does **not** require training any new models. The only model-training that happens is whatever the *evaluation agent* (i.e., Codex in our paper) chooses to do while constructing its solution. The novelty evaluation uses only:
>
> * the agent-generated code and configuration,
> * the resulting textual profile
> * LLM-based comparisons to $S_{\text{known}}$.
>
> By the way, we do not forbid an evaluation agent from training models, but in our experiments, we do not prompt the evaluator to perform any training or optimization.

---

> ### Author Response · Authors · 2025-11-21
> **Response to Reviewer ELHx (Part 2/3, Weakness 2)**
>
> ## 2. Response to W2: Missing technical details (distance function, task processing)
> > **w2:** Several key details are missing in the main text of the paper which can weaken its clarity and reproducibility, for example, what exactly are the distance functions used for each task (seems to be B.2 COMPARISON PROMPT but not mentioned in main text), how are the tasks processed to be use for creativity measure (e.g. some of the tasks are classification based) and meaning of the three methods used in the experiments (e.g. MLAB, CodeAct, AIDE).
>
> We appreciate the reviewer pointing out that some important details were only in the appendix in the previous version. We have revised the main text to be more self-contained.
>
> ### 2.1 Distance function used in our experiments.
> * **Clarification in Main Text:** We clarified in **Section 2** and **Section 4.1** that the Distance function $D$ is instantiated using a two-stage **Agent-as-Judge** pipeline.
> * **Detailed Mechanism:** The specific mechanism is fully detailed in **Appendix E.1** "Recap: Novelty Evaluation Pipeline". This involves:
>     1.  **Extraction (Appx. G.1):** Using an agent named Codex to distill a solution's core strategy into **standardized text** and **pseudocode** representations.
>     2.  **Comparison (Appx. G.2):** Using the **Comparison Prompt** (the rubric mentioned in Appendix G) to compare two solutions along six fixed methodological dimensions, scored on a 0-4 scale.
>     3.  **Formal Mapping:** The final $D_{AGENT}$ score is the normalized average of these rubric scores, formally defined in **Eq. (4)**.
>
> ### 2.2 Task Construction Process
> For each existing benchmark, including classification tasks, we wrap the original competition or dataset into the unified formalism $\mathcal{T} = (P,S,V,D)$:
>
> * **Problem $P$.**
>   $P$ contains the full problem specification: the task description, visible data (e.g., train/validation splits), hidden evaluation data, and the official evaluation script or metric. For classification tasks (e.g., BEETL-MI, BEETL-Sleep, RCIC), $P_{\text{visible}}$ includes only the training/validation sets and interface, while test labels remain in $P_{\text{hidden}}$.
>
> * **Solution space $S$.**
>   $S$ is the space of **executable pipelines** the agent can construct in iGym, not just prediction files. For a classification task, a solution $s \in S$ is typically a inference code: it loads the data and a model, and then writes predictions in the required format.
>
> * **Value function (V(s)).**
>   We compute $V(s) = C(s)\cdot R(s),$ where $C(s)$ is a feasibility check (the pipeline runs successfully and produces a valid submission), and $R(s)$ is the task’s **official metric** (accuracy, F1, AUC, RMSE, etc.), exactly as defined in the original benchmark or competition.
>
> * **Distance $D$.**
>   In all experiments, $D$ is instantiated as the same Agent-as-judge distance $D_{\text{AGENT}}$, which operates on method profiles derived from the solution code (see  Appx. E). And we can add a concrete rubric to evaluate $D$, injecting prior knowledge.
>
> This wrapping is applied uniformly across tasks: classification, recommendation, and optimization benchmarks all become problem instances with executable solution pipelines and a standardized $V$ and $D$.
>
> ### 2.3 Novelty Evaluation Process
> A key point is that our novelty measure does **not** compare raw prediction outputs. The performance is measured by $G$ in Section 2.1. And novelty compares **methods**:
>
> * For any task type, including classification, we first use an **extraction prompt** (see Appendix G.1) to convert the agent’s solution $s$ and each reference solution in $S_{\text{known}}$ into a standardized **method profile**: a natural-language summary plus pseudocode capturing architecture choices, objectives, data usage, training strategy, etc.
>
> * We then use a **comparison prompt** (see Appendix G.2) to score methodological differences between two profiles along six rubric dimensions on a 0–4 scale. Appendix E.1 defines how these discrete scores are aggregated into a continuous distance $D_{\text{AGENT}}(s, h) \in [0, 100]$, and novelty is computed as $$N(s) = C(s)\cdot \min_{h \in S_{\text{known}}} D_{\text{AGENT}}(s, h).$$
>
> Thus, even for a classification task, novelty is measured in terms of **how the method is designed** (model family, loss, data pipeline, optimization, ensembling, etc.), not merely whether its predictions differ numerically. Two solutions with similar accuracy can receive very different novelty scores if one is just a small hyperparameter tweak of existing baselines, while the other uses a substantially different modeling or training approach.

---

> ### Author Response · Authors · 2025-11-21
> **Response to Reviewer ELHx (Part 3/3, Weakness 2)**
>
> ## 2. Response to W2: Missing technical details (baselines)
> This is a continuation of our response to Weakness 2 because of the 5000-character limit.
>
> ### 2.4 Meaning of the three methods: MLAB, CodeAct, AIDE
>
> We agree that in the earlier version, the main text did not sufficiently explain the three baselines. We have added a direct summary of the key methodological differences of the three representative agent frameworks in **Section 4.1** and expanded the details in **Appendix D.1**:
>
> | Agent Framework | Paradigm | Core Mechanism |
> | :--- | :--- | :--- |
> | **MLAB** | Workflow-Style / ReAct | Plans in natural language, issues high-level actions (read/edit files), relies on explicit workflow planning. |
> | **CODEACT** | Code-Centric / Autonomous | Unifies all actions into executable Python code, relying on self-debugging and repeated code execution. |
> | **AIDE** | Tree-Search Optimization | Views the task as code optimization, using tree search to iteratively propose and refine strong configurations. |
>
> ---
>
> Once again, we thank you for the thoughtful and constructive feedback. We believe the clarifications, additional experiments, and textual revisions described above substantially strengthen the paper and make our design choices and limitations more transparent.
>
> **Please let us know if you have any further questions. If you find that our response addresses some of your concerns, would you kindly consider raising your rating score for our paper? We greatly appreciate your consideration.**
>
> ---
>
> Best Regards,
>
> Authors of InnoGym

---

> ### Author Response · Authors · 2025-11-27
> **Kindly Reminder**
>
> Dear reviewer ELHx,
>
> We highly appreciate the constructive comments and insightful suggestions you have offered for our work. As the discussion period has only a few days left, in order for us to have sufficient time to address any additional questions you may have, we kindly encourage you to engage in the ongoing discussion and share any further insights or clarifications you may have.
>
> **If you find that our response addresses some of your concerns, would you kindly consider raising your rating score for our paper?**
>
> Thank you very much for your time and consideration. We look forward to hearing from you soon.
>
> Best Regards,
>
> Authors of InnoGym

---

### Official Review · Reviewer_kLJD · 2025-10-31

**Soundness:** 2
**Presentation:** 1
**Contribution:** 2
**Rating:** 4
**Confidence:** 3

**Summary:**

This paper introduces InnoGym, a benchmark with 18 tasks from real-world engineering and scientific domains to evaluate AI agents on both performance and solution novelty. A core contribution is its novelty metric, which quantifies methodological differences from prior approaches. The paper also tested three popular agents in the community with their proposed benchmark.

**Strengths:**

- Addresses an important and underexplored aspect: evaluating the novelty of agent-generated solutions, going beyond mere correctness.
- Constructs a benchmark with 18 real-world "improvable" tasks.
- Provides iGym, a unified and reproducible execution environment that supports long-horizon agent workflows and could benefit the broader research community (if open-sourced).
- Conducts systematic evaluation of three popular agent frameworks (MLAB, CodeAct, AIDE), offering concrete insights into current capabilities and limitations.

**Weaknesses:**

- **Novelty metric reliability**: The embedding distances of text are highly sensitive to presentation style—identical solutions with different implementations could yield large distances. Conversely, highly novel ideas (e.g., dropout, residual connections) may appear minor in textual or architectural distance but represent significant conceptual leaps. These are known challenges in novelty estimation; it is not reasonable to expect that this paper solves them completely, but the paper should discuss them and clarify the authors’ design choices.

- **Heavy dependence on S_known**: Novelty scores are inherently limited by the completeness and diversity of collected prior solutions. A method may appear highly novel due to missing references. The paper does not address how sensitive results are to its coverage.

- **Lack of methodological detail**:
  - The distance function is not clearly defined in Section 2.1. While N(s) = ... D(s, h) is stated, experiments (Section 4.1) reveal that novelty is computed via LLM-based profiling and scoring across six dimensions (0–4 scale), not raw distance. Also, what exactly distance function used here?
  - **Validator construction**: It is unclear whether validators are rule-based, script-based, or LLM-powered. How are complex constraints (e.g., format, feasibility, domain-specific rules) reliably checked?
  - **Evaluator normalization**: The process of converting relative scores to absolute ones and “adjusting until leaderboard consistency” is confusing. What adjustments were made? Were they manual or automated? How is consistency verified across languages and environments?

- **Agent selection and representativeness**:
  - The three evaluated frameworks (MLAB, CodeAct, AIDE) lack citations.
  - Many benchmark tasks are optimization-focused, yet the selected agents are not known to specialize in optimization. Stronger baselines like OpenEvolve or other evolutionary/optimization-oriented agents could be considered.

**Questions:**

see above weakness

---

> ### Author Response · Authors · 2025-11-21
> **Response to Reviewer kLJD (Part 1/4, Weakness 1)**
>
> Dear Reviewer kLJD,
>
> We thank you for appreciating the strengths of our work, specifically: **the novelty evaluation beyond correctness**, **the 18-task real-world benchmark**, **the iGym unified environment**, and **the systematic study of three agent frameworks**.
>
> A revised draft of our manuscript has been uploaded, with added/modified content marked in green. Below, we respond to each concern in turn.
>
> ## 1. W1: Novelty metric reliability
> > W1: Novelty metric reliability: The embedding distances of text are highly sensitive to presentation style—identical solutions with different implementations could yield large distances. Conversely, highly novel ideas (e.g., dropout, residual connections) may appear minor in textual or architectural distance but represent significant conceptual leaps. These are known challenges in novelty estimation; it is not reasonable to expect that this paper solves them completely, but the paper should discuss them and clarify the authors’ design choices.
>
> **Summary.** Our novelty metric does not use raw text embeddings. Instead, we instantiate $D$ as an **agent-as-judge** metric designed to capture *method-level* differences rather than surface style. We validate this metric on triplet datasets with human judgments, and we explicitly clarify its scope and limitations in the paper.
>
> ### 1.1 Clarifying the distance function $D$: Agent-as-judge instead of embeddings
>
> We fully agree that naive embedding-based distances are too sensitive to superficial style. In our work, **we do not use embedding distances to compute $D$**.
>
> In the revised paper, we clarify in **Section 2.1**, **Section 4.1**, and **Appendix E.1**: in our experiments, $D$ is instantiated as an **Agent-as-judge** metric using Codex (with gpt-5-high as backbone), implemented in two stages:
>
> 1. **Extraction stage**
>
>    * For each solution, we use the prompt in **Appendix G.1** to ask Codex to produce:
>
>      * a high-level **summary.md** of the solution
>      * a **pseudocode.tex** version of the core algorithm.
>    * This step intentionally removes variable names, code layout, and other stylistic details, and keeps only the core strategy and workflow.
>
> 2. **Comparison stage**
>
>    * For an agent solution and a reference solution from $S_{\text{known}}$, we give the extracted summaries and pseudocode.
>    * Using the rubric prompt in **Appendix G.2**, Codex scores the pair along **six methodological dimensions** (e.g., problem formulation, method/algorithm, architecture/implementation, data and features, optimization, evaluation), each on a **0–4** scale (0 = almost identical, 4 = completely different).
>    * We then normalize and average these six scores to obtain $D(s,h)$.
>
> The exact formula and pseudocode for this Agent-as-judge distance are now given in **Section 4.1** and **Appendix E.1**. This design focuses $D$ on *method-level differences* rather than surface presentation. In addition, this method allows us to construct specific rubrics (i.e., evaluation criteria) for different tasks, introducing priors.
>
> ### 1.2 Validating $D$: EquiBench[1] and human triplet experiments
>
> To address reliability, we added experiments in **Appendix E** to test whether our $D$ aligns with human judgments.
>
> * We collected **50 triplets** $(A,B,C)$ from **EquiBench**[1] and created **3 additional triplets** manually. In each triplet, by construction:
>   * $A$ and $B$ are **more similar** in method,
>   * $A$ and $C$ are **more different**.
>     So we expect $D(A,C) > D(A,B)$.
> * We then **sampled 11 triplets** and asked human annotators to judge which pair is more different, and compared these judgments against our Agent-as-judge distances.
>
> The table below shows the correlation scores with human judgments, corresponding to Table 10 in the paper.
>   | Dataset | # Triplets | Pearson r (A,B) | Pearson r (A,C) | Spearman ρ (A,B) | Spearman ρ (A,C) | Triplet agreement |
> |---|---:|---:|---:|---:|---:|---|
> | EquiBench (annot.) | 8         | n/a             | 0.84            | n/a              | 0.87             | 6/8 = 75%         |
> | Human triplets   | 3          | 1.00            | 0.99            | 1.00             | 1.00             | 3/3 = 100%        |
>
> Other results are reported in **Appendix E (Table 8, Table 9, Table 10, and Table 12)**. We find:
>
> * For the vast majority of triplets, our metric satisfies $D(A,C) > D(A,B)$, matching the known construction.
> * On the randomly sampled subset, the Agent-as-judge distances show good correlation and agreement with human judgments.
>
> These results support that, at the granularity relevant to our benchmark, **our instantiation of $D$ is a reasonable and stable proxy for measuring “method-level difference to $S_{\text{known}}$”**. Because we do not judge the method solely by its text description, pseudocode that was extracted by an agent is also included. We also explicitly state in the Limitations that this is one concrete instantiation of $D$, not a final solution to novelty estimation in general.

---

> > ### Author Response · Authors · 2025-11-21
> > **Response to Reviewer kLJD (Part 2/4, Weakness 2)**
> >
> > ## 2. Response to W2: Heavy dependence on $S_{\text{known}}$
> >
> > > W2: Heavy dependence on S_known: Novelty scores are inherently limited by the completeness and diversity of collected prior solutions. A method may appear highly novel due to missing references. The paper does not address how sensitive results are to its coverage.
> >
> > **Summary.** We describe in detail how $S_{\text{known}}$ is constructed from multiple sources and multiple searchers, clarify that $N(s)$ is defined *relative to this set* and can be viewed as an upper bound under the current coverage, and explicitly discuss the unavoidable limitations of incomplete coverage and possible future improvements.
> >
> > ### 2.1 Construct $S_{\text{known}}$
> >
> > We agree that novelty scores are conceptually tied to the set of known solutions. In the revision, **Appendix F.1** now describes the construction of $S_{\text{known}}$ in detail. In short:
> >
> > 1. **Multi-source collection.**
> >    For each task, we gather candidate solutions from:
> >
> >    * official **leaderboards** and competition reports,
> >    * **papers** describing top solutions or follow-up work,
> >    * public **code repositories** (e.g., GitHub, Kaggle notebooks),
> >      using Google Scholar and ChatGPT to broaden the search.
> >
> > 2. **Multiple independent searchers.**
> >    Several authors independently search and collect candidate solutions for each task, then merge their findings. This reduces dependence on any single person’s prior knowledge.
> >
> > 3. **Reproducibility filtering**
> >    All candidate solutions are run in a unified environment. Only those that execute successfully and reproduce the reported performance (or are very close) are retained in $S_{\text{known}}$.
> >
> > We also now report, in Table 3, the **number of reference solutions per task** and the **diversity of reference solutions**. As we note, many of our tasks are **narrow, competition-style problems**, which naturally have fewer independently developed public solutions than very broad tasks like “image classification”.
> >
> > ### 2.2 Dynamic $S_{\text{known}}$ and the “upper bound” view of $N(s)$
> >
> > In **Section 2.2**, we define:
> > $N(s) = C(s)\cdot \min_{h\in S_{\text{known}}} D(s,h)$
> >
> > We emphasize:
> > * $S_{\text{known}}$ is **not fixed** forever. Once the benchmark is public, we expect the community and domain experts to help expand it over time by contributing new verified solutions.
> > * With this definition, enlarging $S_{\text{known}}$ can only **keep $N(s)$ the same or reduce it**, but never increase it.
> > * Therefore, the novelty scores reported in this paper can be interpreted as an **upper bound on novelty with respect to the current $S_{\text{known}}$**.
> >
> > ### 2.3 Fairness and the *relative* nature of novelty
> >
> > We also clarify:
> >
> > * For a given task, **all agents are evaluated against the same $S_{\text{known}}$**. If a new high-quality solution is later added to $S_{\text{known}}$, this affects all future evaluations for that task in the same way.
> > * Our main use of $N(s)$ is **comparative**: we are primarily interested in *how different agents’ solutions differ from known methods under the same $S_{\text{known}}$*, not in assigning an absolute, task-independent novelty score.
> >
> > Thus, even though coverage is not perfect, the **relative comparisons between agents on the same task are still meaningful and fair**.
> >
> > ### 2.4 Difficulty of complete coverage and human limitations
> >
> > We explicitly acknowledge that **complete coverage is impossible** in practice:
> >
> > * No process—manual or automated—can exhaustively enumerate all relevant methods for a non-trivial task.
> > * Human experts themselves face the same limitation: when they evaluate whether a new paper or algorithm is “novel”, they do so under their **own partial knowledge and subjective priors**.
> >
> > In our setup:
> >
> > * We try to minimize omissions by combining multiple sources (leaderboards, papers, code, search) and multiple human searchers.
> > * If a method is so rare that it hardly appears in public literature or code, and an agent discovers such a method and achieves strong performance gains $G$, **treating it as novel is arguably reasonable in practice**—much like how a human reviewer would think “this looks new given what I know”.
> >
> > We state this limitation explicitly, to make clear that some subjectivity and incompleteness are inherent to novelty assessment, whether done by humans or by our metric.
> >
> > ### 2.5 Future directions: using specialized agents to improve $S_{\text{known}}$ and $D$
> >
> > We agree that there is room to make $S_{\text{known}}$ and $D$ more “intelligent”. Here, we outline a direction for **future work**:
> >
> > * Design a **specialized agent** whose job is to:
> >
> >   * search the literature and code repositories,
> >   * construct and update $S_{\text{known}}$,
> >   * and compare new solutions against this enlarged knowledge base.
> > * This would give a more powerful instantiation of $D$, still within our general framework.

---

> > > ### Author Response · Authors · 2025-11-21
> > > **Response to Reviewer kLJD (Part 3/4, Weakness 3)**
> > >
> > > ## 3. Response to W3: Methodological details
> > >
> > > > W3: Lack of methodological detail:
> > > > **W3.1**: The distance function is not clearly defined in Section 2.1. While N(s) = ... D(s, h) is stated, experiments (Section 4.1) reveal that novelty is computed via LLM-based profiling and scoring across six dimensions (0–4 scale), not raw distance. Also, what exactly distance function used here?
> > > > **W3.2**: Validator construction: It is unclear whether validators are rule-based, script-based, or LLM-powered. How are complex constraints (e.g., format, feasibility, domain-specific rules) reliably checked?
> > > > **W3.3**: Evaluator normalization: The process of converting relative scores to absolute ones and “adjusting until leaderboard consistency” is confusing. What adjustments were made? Were they manual or automated? How is consistency verified across languages and environments?
> > >
> > > **Summary.** We clarify the implementation of $D$, provide detailed descriptions of validators (which are purely rule-based/scripted), and precisely explain the evaluator normalization procedure, emphasizing that all adjustments are global, automated, and monotone—not manual tweaks of agent scores.
> > >
> > > ### 3.1 Response to W3.1: Distance function $D$
> > >
> > > As discussed under W1, in the revision:
> > >
> > > * **Section 2.1** defines $D$ abstractly, and explicitly states that in this paper we instantiate it via an Agent-as-judge approach.
> > > * **Section 4.1 and Appendix E.1** then give the full implementation:
> > >
> > >   * extraction of summaries and pseudocode (Appendix G.1),
> > >   * six-dimensional 0–4 scoring (Appendix G.2),
> > >   * and the normalization that turns these scores into $D(s,h)$ and $N(s)$.
> > >
> > > ### 3.2 Response to W3.2: Validator construction
> > >
> > > We agree that the validator design was under-specified. In the revision, **Appendix F.3** describes the validators in detail, and we summarize the key points here:
> > >
> > > * **Validators do not use LLMs.** All validators are manually constructed scripts or rule-based checks derived from each task’s specification.
> > > * We support two main types of submissions:
> > >
> > >   1. **Code submissions**
> > >
> > >      * The validator checks that the required functions or entry points are defined,
> > >      * runs the code on a small fixed input,
> > >      * and verifies that it executes without crashing and returns outputs of the correct type/shape.
> > >   2. **Answer-file submissions**
> > >
> > >      * The validator checks file format (e.g., CSV/JSON), column/field names,
> > >      * and verifies that values lie in the expected domains (e.g., valid labels, numeric ranges, no missing required fields).
> > >
> > > All validators are **deterministic**: the same submission always gets the same pass/fail result, which is crucial for debugging and reproducibility.
> > >
> > > ### 3.3 Response to W3.3: Evaluator normalization
> > >
> > > We also clarify the evaluator normalization in **Appendix F.2**. The main steps are:
> > >
> > > * For competitions that report **relative scores** (e.g., ranking, percentiles), we reconstruct **absolute scores** by re-running the **official evaluation scripts** on the original data, using the available public submissions.
> > > * We then apply a **monotonic transformation** (such as log normalization) to map scores to a shared range.
> > > * We check that the reconstructed scores are consistent with the original leaderboard (e.g., via rank correlation).
> > > * For evaluators implemented in languages like C++ or Java, we **containerize** them (e.g., via Docker) and expose a simple Python wrapper, so that scoring is uniform across languages and platforms.
> > >
> > > Crucially:
> > >
> > > * **All adjustments are done at the evaluator level**, are **global and monotone**, and are applied uniformly to all submissions.
> > > * We **never manually tweak individual agent scores**.
> > >
> > > Once an evaluator is fixed:
> > >
> > > * **All agent scores are produced automatically by the evaluator.**
> > > * We **do not hand-tune or manually adjust any agent’s score**.

---

> > > > ### Author Response · Authors · 2025-11-21
> > > > **Response to Reviewer kLJD (Part 4/4, Weakness 4)**
> > > >
> > > > ## 4. Response to W4: Agent selection and representativeness
> > > >
> > > > > W4: Agent selection and representativeness:
> > > > > W4.1: The three evaluated frameworks (MLAB, CodeAct, AIDE) lack citations.
> > > > > W4.2: Many benchmark tasks are optimization-focused, yet the selected agents are not known to specialize in optimization. Stronger baselines like OpenEvolve or other evolutionary/optimization-oriented agents could be considered.
> > > >
> > > > **Summary.** We add citations and descriptions for the evaluated frameworks, explain our focus on widely-used general-purpose agent frameworks in this first study, and position specialized optimization/evolutionary agents such as OpenEvolve as an important next step that our released benchmark and iGym infrastructure are designed to support.
> > > >
> > > > ### 4.1 Response to W4.1: Lack Citations
> > > >
> > > > In the revised paper, **Section 4.1** now includes proper citations for: **MLAB**, **CodeAct**, and **AIDE**.
> > > >
> > > > We also add short descriptions (and more details in Appendix D.1) explaining how each framework works. This aligns our setup with prior agent benchmarks such as MLE-Bench.
> > > >
> > > > ### 4.2 Response to W4.2: Include OpenEvolve-style agents
> > > >
> > > > We agree that evolutionary/optimization-oriented agents like **OpenEvolve** are strong and very relevant baselines for many of our tasks. We did not include them in this version for the following reasons:
> > > >
> > > > 1. **Dependence on a non-trivial initial program and different problem setting**
> > > >
> > > >    * OpenEvolve-style methods usually assume a fairly substantial **initial program** (e.g., `init_program.py`) and evolve from there.
> > > >    * For many iBench tasks, it is hard to provide a single, simple, and effective initial codebase that fits our “no reference solution is given” setting.
> > > >    * If we used solutions in (S_{\text{known}}) as the initial program, we would be evaluating “innovation *given a reference solution*”, which is a different setting from the one we study in this paper.
> > > >
> > > > 2. **There is an agent that has strong optimization capabilities.**
> > > >
> > > >    * The primary focus of this paper is to **propose the G+N framework and build the InnoGym / iBench / iGym infrastructure**, and to demonstrate its use with three representative agent frameworks that are already widely used in prior work.
> > > >    * Some of the evaluated frameworks (e.g., AIDE) already have strong **automatic optimization capabilities**, partially covering the “optimization agent” axis.
> > > >
> > > >
> > > > We state that integrating **specialized optimization/evolutionary agents** such as OpenEvolve into iGym and evaluating them on iBench is an important next step. All tasks, evaluators, and the iGym will be released to facilitate such follow-up work.
> > > >
> > > > ---
> > > >
> > > > Once again, we thank you for the thoughtful and constructive feedback. We believe the clarifications, additional experiments, and textual revisions described above substantially strengthen the paper and make our design choices and limitations more transparent.
> > > >
> > > > **Please let us know if you have any further questions. If you find that our response addresses some of your concerns, would you kindly consider raising your rating score for our paper? We greatly appreciate your consideration.**
> > > >
> > > > ---
> > > >
> > > > ## Reference
> > > > [1] EquiBench. ACL2025. https://arxiv.org/abs/2502.12466.
> > > >
> > > > ---
> > > >
> > > > Best Regards,
> > > >
> > > > Authors of InnoGym

---

> > > > > ### Comment · Reviewer_kLJD · 2025-11-26
> > > > >
> > > > > I thank the authors for their detailed response and insightful discussion on my comments. Most of my concerns are now addressed. I will increase my score to support the acceptance of this paper.

---

> > > > > > ### Author Response · Authors · 2025-11-26
> > > > > > **Thank you for your support and increasing the score**
> > > > > >
> > > > > > Dear Reviewer kLJD,
> > > > > >
> > > > > > We are sincerely grateful for your positive feedback and your decision to increase the score. We appreciate the time you took to review our rebuttal. We will ensure that all the discussed clarifications and improvements are incorporated into the final version of the paper.
> > > > > >
> > > > > > Best Regards,
> > > > > >
> > > > > > Authors of InnoGym

---

### Author Response · Authors · 2025-11-21
**General Response to All Reviewers: Summary of Revisions in Paper for Each Reviewer**

Dear all reviewers,

Thank you for your thoughtful reviews!

We sincerely appreciate the reviewers’ feedback and questions, which have helped us improve the clarity and completeness of the manuscript. **We emphasize that the core methodology and main experimental conclusions remain unchanged. Most revisions are confined to the Appendix, while changes in the main text are minimal and purely for clarification and readability.** A revised draft of the manuscript has been uploaded, with added or modified content highlighted in green.

These revisions primarily focus on clarifying dataset construction details and refining the Novelty evaluation protocol.

**For Reviewer kLJD:**

- Clarified that the distance function $D$ uses an **Agent-as-Judge** approach rather than embeddings. (Page 3, Lines 113-115)
- Added citations for the three Agent frameworks. (Page 6, Line 294)
- Detailed the implementation method of the distance function $D$. (Page 6, Lines 301-316; Page 20, Appx. E.1)
- Added a detailed description of the experimental setup. (Page 6, Lines 318-323)
- Included statistics on the size and diversity of the known solution set $S_{known}$ for each task. (Page 16, Appx. C and Tab. 3)
- Conducted experiments comparing the distance function $D$ with human evaluation. (Page 20, Appx. E.2-E.4 and Tab. 9, 10, 11, 12)
- Added a detailed description of **Solution Collection**. (Page 22, Appx. F.1)
- Added a detailed description of **Evaluator Normalization**. (Page 24, Appx. F.2)
- Added a detailed description of **Validator Construction**. (Page 25, Appx. F.3)

**For Reviewer ELHx:**

- Clarified that the distance function $D$ uses an **Agent-as-Judge** approach rather than embeddings. (Page 3, Lines 113-115)
- Detailed the implementation method of the distance function $D$. (Page 6, Lines 301-316; Page 20, Appx. E.1)
- Added citations for the three Agent frameworks. (Page 6, Line 294)
- Added an overview of the three Agent frameworks. (Page 17, Appx. D.1)

**For Reviewer Rk9e:**

- Added citations for the three Agent frameworks. (Page 6, Line 294)
- Detailed the implementation method of the distance function $D$. (Page 6, Lines 301-316; Page 20, Appx. E.1)
- Added a detailed description of the experimental setup. (Page 6, Lines 318-323)
- Included statistics on the size and diversity of the known solution set $S_{known}$ for each task. (Page 16, Appx. C and Tab. 3)
- Added a detailed explanation of the **Task Taxonomy**. (Page 16, Appx. C.1)
- Added an overview of the three Agent frameworks. (Page 17, Appx. D.1)
- Added experiments on explicitly prompting agents for innovation. (Page 19, Appx. D.3, Lines 1017-1023 and Tab. 7)
- Conducted experiments comparing the distance function $D$ with human evaluation. (Page 20, Appx. E.2-E.4 and Tab. 9, 10, 11, 12)
- Added a detailed description of **Solution Collection**. (Page 22, Appx. F.1)

**For Reviewer UjRg:**

- Corrected the definition of formula $V$. (Page 3, Line 120)
- Detailed the implementation method of the distance function $D$. (Page 6, Lines 301-316; Page 20, Appx. E.1)
- Added a detailed description of the experimental setup. (Page 6, Lines 318-323)
- Added error bars to Figure 5. (Page 8, Fig. 5)
- Added a detailed explanation of the **Task Taxonomy**. (Page 16, Appx. C.1)
- Included statistics on the size and diversity of the known solution set $S_{known}$ for each task. (Page 16, Appx. C and Tab. 3)
- Added statistical analysis for Table 2. (Page 18, Appx. D.2 and Tab. 4, 5)
- Added submission success rates for the main experiments. (Page 18, Appx. D.3 and Tab. 6)
- Conducted experiments comparing the distance function $D$ with human evaluation. (Page 20, Appx. E.2-E.4 and Tab. 9, 10, 11, 12)
- Added a detailed description of Solution Collection. (Page 22, Appx. F.1)

Best Regards,
Authors of InnoGym

---

### Author Response · Authors · 2025-11-22
**General Response to All Reviewers: Summary of Strengths**

**Dear All Reviewers,**

We sincerely thank you for your time and the insightful and constructive feedback on our paper. We are greatly encouraged by the positive reception from all reviewers regarding the significance of our topic, the utility of our proposed benchmark and environment, and the rigor of our evaluation.

We summarize the **common strengths** acknowledged by the reviewers as follows:

* **Important and underexplored topic:** **All reviewers** acknowledged the significance of evaluating the novelty of agent-generated solutions. **kLJD** highlighted that the paper addresses an "important and underexplored aspect" that goes "beyond mere correctness." **Rk9e** found the evaluation of novelty "interesting and important," noting that the quantification is well-grounded. **ELHx** further confirmed that our approach to evaluating creativity is "sound and intuitive."

* **High-quality benchmark and dataset:** Reviewers commended the construction and curation of our benchmark.
    * **kLJD** noted the value of constructing a benchmark with `18 real-world 'improvable' tasks.`
    * **Rk9e** praised the `good comparison with existing benchmarks` (Table 1), which helps consolidate the motivation.
    * **UjRg** emphasized that the dataset curation is `well-documented,` specifically appreciating the "two-stage filtering" and the "clear visible/hidden split of artifacts."
    * **ELHx** added that the suite of tasks could be a `helpful resource for the AI community.`

* **Unified and reproducible environment (iGym):** The proposed execution environment received strong endorsement for its utility and design.
    * **kLJD** described iGym as a `unified and reproducible execution environment` that supports "long-horizon agent workflows."
    * **UjRg** praised the `great abstractions` in the iGym SDK (e.g., recovery+concurrency) and noted that the `formalization and metrics are "clear" and useful for "measuring breakthroughs."`
    * **ELHx** agreed that the gym is a valuable resource for the community.

* **Systematic evaluation and insightful analysis:** Reviewers appreciated the depth of our empirical work.
    * **kLJD** commended the `systematic evaluation` of three popular agent frameworks (MLAB, CodeAct, AIDE), stating it offers `concrete insights into current capabilities.`
    * **Rk9e** found the analysis, particularly regarding prior solutions, to be `well-executed.`
    * **UjRg** also noted that the feasibility gating ensures novelty is credited only to valid solutions.

* **Clarity of presentation:** Reviewer **Rk9e** noted that the paper is `well-written and easy to follow,` with `clear figurative illustrations.`

In the individual responses below, we have carefully addressed the specific questions and suggestions raised by each reviewer. We hope our responses and the revised manuscript satisfactorily address your concerns.

Best regards,

Authors of InnoGym

---

### Author Response · Authors · 2025-12-03
**Summary for Area Chairs (Part 1/2, Rebuttal Engagement, Motivation, and Contributions)**

Dear (Senior) Area Chairs and Program Chairs,

Thank you for handling our paper and for coordinating the review process. We are grateful to the reviewers for their time and constructive feedback. Below, we summarize the rebuttal engagement, restate the motivation and contributions, highlight strengths noted by the reviewers, and concisely summarize how the revision addresses the main concerns.


## Rebuttal Engagement
We provided detailed responses to all reviewers. Only reviewer kLJD engaged during the rebuttal, indicated that our clarifications addressed their concerns, and raised the score accordingly. We invite you to consult the full rebuttal (both the overall response and the per-reviewer replies) and the marked-up revision (**changes highlighted in green**). We believe the paper now addresses all concerns while preserving and strengthening its original contributions.

## Motivation and Contributions
Existing benchmarks for LLMs and agents primarily measure **Correctness**, focusing on whether an output passes test cases or matches a reference answer. However, true scientific and engineering innovation depends not only on results but also on **Methodological Diversity**. Under the current paradigm, two agents arriving at the same correct answer via fundamentally different approaches are treated identically, causing us to overlook the originality in the problem-solving process.

We introduce **InnoGym**, the first benchmark and framework designed to systematically evaluate the innovation potential of AI agents. We aim to bridge the gap between mere performance evaluation and abstract creativity generation.

**Concretely, InnoGym offers the following key contributions:**

1. **A Principled Innovation Evaluation Framework:** We formalize tasks as a quadruple $(P, S, V, D)$ and introduce two complementary metrics: **Performance Gain ($G$)** to measure improvement over the best-known solutions, and **Novelty $N$** to quantify the difference from prior approaches. Crucially, we introduce **Feasibility Gating $C(s)$**, ensuring that novelty is credited only to valid solutions.
2. **The InnoGym Benchmark (iBench):** We meticulously curated **18 standardized tasks** derived from real-world scientific and engineering competitions (e.g., NeurIPS, KDD Cup). These are defined as **"Improvable Tasks"**—distinct from "Solved Problems" (known optimum) and "Exploratory Problems" (no validation)—providing agents with the space to demonstrate both performance breakthroughs and methodological innovation.
3. **A Unified Execution Environment (iGym):** To support long-horizon reasoning and complex tool use, we developed iGym. It provides a unified abstraction layer, native concurrency support, and **Recovery** mechanisms for long-running tasks, ensuring reproducible comparisons across different agent frameworks (e.g., MLAB, CodeAct, AIDE) under consistent conditions.

Building on InnoGym, we conducted extensive experiments on different agents. Our findings reveal that while some agents (e.g., MLAB) demonstrate high novelty, current agents often lack **Robustness** on complex tasks, preventing high novelty from translating into practical performance gains. **This highlights a critical gap between "creativity" and "effectiveness" in current AI agents. We will release the iBench dataset and the iGym environment to support the community's future research on agentic innovation.**

---

### Author Response · Authors · 2025-12-03
**Summary for Area Chairs (Part 2/2, Strengths and Addressed Main Concerns)**

## Strengths
- **Important and underexplored topic** (all reviewers).

- **High-quality benchmark and dataset** (all reviewers).

- **Unified, reproducible environment that benefits the community** (Reviewer kLJD, UjRg, and ELHx).

- **Systematic evaluation and insightful analysis** (Reviewer kLJD, Rk9e, and UjRg).


## Addressed Main Concerns

1. **Distance function $D$ (kLJD`#1` `#3`; ELHx`#2`; Rk9e`#2`; UjRg`#1` `#4`)**

    In Sections 2.1 and 4.1, we add a concise description of how $D$ is instantiated via an **agent-as-judge** evaluator: we first extract a method/profile and pseudocode for each solution, then compare against known solutions using a six-dimension 0–4 rubric and normalize to a 0–100 score $D_{\text{AGENT}}$. **Appendix E.1** provides the formalized implementation, ensuring §2.1 and §4.1 are fully consistent.

2. **$S_{\text{known}}$ coverage and potential bias (kLJD`#2`; UjRg`#5`)**

    In **Appendix F.3**, we added details regarding the collection process of $S_{\text{known}}$, which adopts an AI-assisted, multi-searcher independent retrieval and merging method. This design demonstrates the completeness of the collection process. Then we report per task, the number of collected solutions and diversity statistics in **Table 3 of Appendix C**. We clarify that full coverage is difficult even for human experts, so $S_{\text{known}}$ can evolve and we invite community contributions. On **fairness**, we state in the **rebuttal** that all frameworks are compared under the **same fixed $S_{\text{known}}$**; since $N(s)=\min_{h\in S_{\text{known}}}D(s,h)$, enlarging $S_{\text{known}}$ can only keep or **reduce** $N$, not inflate it.

3. **Novelty accuracy and subjectivity (Agent-as-judge) (kLJD`#1`; Rk9e`#3`; UjRg`#1`)**

    We validate on **EquiBench** (mentioned by reviewer Rk9e) and with a human spot-check set (see **Appendix E.2/E.3**). The metric clearly separates cosmetic vs. algorithmic variants (e.g., mean distances ≈ 1.00 vs. ≈ 9.75) and shows strong agreement with human judgments (e.g., Pearson $r\approx0.84$ on an EquiBench subsample; $r\approx0.99-1.00$ on human triplets). Because EquiBench is constructed via equivalence/perturbations, we **do not** add extra perturbation studies.

4. **Dataset construction details (kLJD`#3`; ELHx`#2`; UjRg`#5`)**

    In **Appendix F.2/F.3**, we document the end-to-end pipeline: task preprocessing → candidate generation → **deterministic** validators (rule/script; no LLMs) → scoring and normalization → cross-language consistency. We also clarify in the main text how the comparison prompt relates to $D$, ensuring reproducibility and auditability.

5. **Statistical analysis and reporting (UjRg`#3`)**

    In **Appendix D.2**, we add significance tests and bootstrap analyses for **Table 2** ( see **Tables 4 and 5 in Appendix** ), showing **MLAB** is significantly better than **AIDE** and **CodeAct** on key metrics. For the §4.3 analysis experiments, we run each setting **three times** and report **error bars**; results remain consistent with the original conclusions.

6. **Role and boundaries of novelty (Rk9e`#4` `#5`)**

    We clarify that **performance** and **novelty** are two independent axes of innovation. Under a **performance constraint**, novelty serves as an operational proxy for **usable diversity**. Our taxonomy and visualizations (e.g., Fig. 4b) help characterize the known solution space and diagnose whether agents **replicate** known methods or **explore** new ones; we do **not** claim downstream “practice gains” beyond this diagnostic role.

---

Thank you again for your time and coordination. We are confident that InnoGym will serve as a valuable asset for the community.

---

Best regards,

Authors of InnoGym

---

### Meta-Review · Area_Chair_KbhY · 2026-01-06

**Summary:**

Reviewers raised several concerns: (1) the definition of “solved problems” is overly restrictive; (2) the novelty metric relies on a limited set of pre-existing solutions and (3) the generalizability of InnoGym to broader or real-world innovation scenarios remains unclear. While they acknowledged the framework’s novelty and thorough experimental validation.

**Reviewer Concerns:**

The rebuttal addressed key concerns: the distance function was clarified; reference solution coverage and bias were mitigated by detailing the AI-assisted collection process; novelty subjectivity was validated via EquiBench and human evaluations, demonstrating strong agreement; dataset construction was made reproducible through deterministic, LLM-free validation; statistical robustness was improved with significance tests and repeated runs; and the role of novelty was clarified as a diagnostic—not a claim of practical impact.

**Reviewer Scores:**

One reviewer states that the score will be improved, maybe from 4 to 6. While other 3 reviewers didn't participate the discussion.

---

### Decision · Program_Chairs · 2026-01-26

Accept (Poster)